# A Possible Land Cover EAGLE Approach to Overcome Remote Sensing Limitations in the Alps Based on Sentinel-1 and Sentinel-2: The Case of Aosta Valley (NW Italy)

**Tommaso Orusa** [1,2,*] **, Duke Cammareri** [2] **and Enrico Borgogno Mondino** [1]

[1] Department of Agricultural, Forest and Food Sciences (DISAFA), GEO4Agri DISAFA Lab, Università degli Studi di Torino, Largo Paolo Braccini 2, 10095 Grugliasco, Italy
[2] Earth Observation Valle d'Aosta—eoVdA, Località L'Île-Blonde 5, 11020 Brissogne, Italy
* Correspondence: tommaso.orusa@gmail.com

**Abstract:** Land cover (LC) maps are crucial to environmental modeling and define sustainable management and planning policies. The development of a land cover mapping continuous service according to the new EAGLE legend criteria has become of great interest to the public sector. In this work, a tentative approach to map land cover overcoming remote sensing (RS) limitations in the mountains according to the newest EAGLE guidelines was proposed. In order to reach this goal, the methodology has been developed in Aosta Valley, NW of Italy, due to its higher degree of geomorphological complexity. Copernicus Sentinel-1 and 2 data were adopted, exploiting the maximum potentialities and limits of both, and processed in Google Earth Engine and SNAP. Due to SAR geometrical distortions, these data were used only to refine the mapping of urban and water surfaces, while for other classes, composite and timeseries filtered and regularized stack from Sentinel-2 were used. GNSS ground truth data were adopted, with training and validation sets. Results showed that K-Nearest-Neighbor and Minimum Distance classification permit maximizing the accuracy and reducing errors. Therefore, a mixed hierarchical approach seems to be the best solution to create LC in mountain areas and strengthen local environmental modeling concerning land cover mapping.

**Keywords:** EAGLE land cover; Sentinel-1 SAR; Sentinel-2; Planetscope; Google Earth Engine; SAGA GIS; Orfeo Toolbox; ESA SNAP; Aosta Valley NW Italy; mountains; mixed hierarchical approach; AI for environmental modeling

## 1. Introduction

The recent massive amount of data from Earth Observation (EO) missions and geospatial cloud-based platforms, coupled with advanced machine learning approaches, are showing a high capability of reducing processing time and enabling effective EO services [1–3]. This mainly relies on the ongoing technological transfer, which is affecting both the research sector and the entrepreneurial one, even supported by significant investments from the spatial economy [4,5]. Nevertheless, the process has not still reached the public sector, especially in alpine and rural areas. This makes it desirable to develop and consolidate proper EO tools for these subjects, thus ensuring higher effectiveness of public institutions while managing ordinary territorial problems [5,6]. This process fits well with the Next Generation UE policy and the Recovery post-SARS-CoV2 pandemic plan that are largely focusing the European economy on investments in digitalization, environmental sustainability, and social inclusion [7]. Earth Observation data may certainly support the reaching of the Sustainable Development Goal, as well as the European one. Nowadays, land cover maps, according to new policies such as EAGLE in the European Union, represent a key point in the green policies, making it possible to assess the effects of both climate change and human pressure on natural resources [8–10].

Nowadays, continued cloud-based land cover services and computer services starting from EO data provide a valid and useful tool in planning and decision-making. Nevertheless, the main critical point that remains is the land cover classification approach adopted that requires updated, validated, and detailed knowledge about a given territory that cannot be reached by a wide global scale classification system [11]. Even if cloud-based services give the possibility of obtaining yearly and newest worldwide land cover map (hereinafter called LCM), especially at the higher geometric resolution, most of them still suffer from bias and errors due to the limits that remote sensing has in geomorphological complexity as well as of a consistent and sufficiently updated number of training sets that often appear statistically not representative for the entire global mountain areas [12]. It is worth noting that also in official products released by space agencies, inaccuracies and errors particularly affect mountain areas, not allowing their applicability for detailed management or planning activities [1]. Products such as Google Dynamic World [13] or ESRI Global Land Cover, or ESA Global Land cover, suffer from these biases in the mountain area [14,15]. The adopted remote sensing approach certainly has its usefulness on a global scale but shows strong inefficiencies in the mountain environment. It is interesting to note that although alpine ecosystems such as the cryosphere are more affected by climate change still today at the level of geoscientific applications and remote sensing, no attempt has been made to develop a solid procedure for the accurate mapping of global mountain land cover [6]. Key ecosystems in terms of adaptation and mitigation if we consider, for example, only the Alpine hydrological component and the central role it plays at a global level. Under this core issue, this work has been developed trying to try to fill this gap. Firstly, it is necessary to discriminate the land use and land cover [16]. Land cover is defined as the observed (bio)physical cover type overlaying Earth's surface, i.e., forests, agricultural areas, human settlements, glaciers, water, and wetlands (see Directive 2007/02 of the European Commission). It is worth reminding that Land cover is not land use and that, in general, EO data can provide LCM and not land use Maps. These can be, conversely, generated from LCM exploiting auxiliary information from other sources [17,18]. Some improvements that move potentialities of EO data from land cover to land use mapping come from the recent augmented temporal resolution of the ongoing higher resolution missions such as Sentinel-2 [19].

The main aim of this work has been to develop a strong approach to map land cover in complex geomorphological areas scalable to all mountain realities going beyond EO data limits and produce land cover maps with the highest accuracy by adopting the most suitable algorithms. In particular, performing a data fusion of both SAR and multispectral EO data exploits the capabilities that both offer in order to reduce the limits that, at the same time, characterize them in an alpine context. This is why a hierarchical approach has been adopted by mapping only certain territorial components with a given sensor or more sensors, avoiding the limits that characterize them in the mountain environment. In this case, land cover maps, according to the European guideline in the EAGLE framework, have been developed due to the study area [20,21]. It is worth noting that the EIONET Action Group on land monitoring in Europe (well known as the EAGLE group) is an open assembly of technical experts from different European Economic Areas (EEA) of the Member States, mostly related to the National Reference Centers (NRC) for land cover mapping. Currently, EAGLE actions are funded by EEA within the framework of the Copernicus program. The Italian NRC is the Istituto Superiore per la Protezione e la Ricerca Ambientale (ISPRA). The distinction between land use and land cover is precisely the core principle of EAGLE. The EAGLE reference tool is a matrix that combines three descriptors: land cover components (LCC), land use attributes (LUA), and extra characteristics (LCH). In order to construct specific categorization systems that are appropriately calibrated for different needs or to find correspondences with already defined class definitions while keeping descriptor independence, the descriptors can be crossed only in certain situations [22].



In order to produce land cover maps, supervised or automatic classification can be performed. Approaches for automatic classification generally take longer as data volume and dimension increase [23]. Additionally, because it is based on an a-posteriori analysis, the interpretation of the results may be inaccurate. This is why the supervised are generally preferred in this type of activity. However, supervised algorithms, which rely on prior knowledge of the classes to map and require a sufficient number of training examples, are more effective when evaluating large amounts of data [24]. They are used to hone the algorithm's ability to forecast class type at undetermined places over images (calibrate model parameters) [25]. Due to the requirement that one sample zone (Area of Interest, AOI) be at least minimally known in terms of type and position over the image, this rendered them weaker in comprehensively mapping all the classes present in the research area. In any case, the abundance, consistency, and correct spatial distribution of the training areas as well as of the validation areas is fundamental, and this, as previously said, is a limiting factor that is often extremely significant on a global level with few areas in mountainous areas. Without considering the need for their continuous updating that only a continuous ground control can offer as regards the development of change detection remote sensing services for land cover [26,27]. Nowadays, the last version of the CORINE Land Cover map (CLC) 2018 ensures a high degree of thematic detail at the European level for different dates. Nevertheless, it is still limited in terms of spatial detail and updating frequency (https://land.copernicus.eu/pan-european/corine-land-cover accessed on 6 November 2022). In the last years, the Higher Resolution Layers (HRLs) from the Copernicus Land Monitoring Service have provided LCM, with reference to the main classes, with greater spatial detail and maintaining a multi-year updating frequency. Unfortunately, these products still show low thematic accuracies in the classification of alpine areas [28]. The CLC product does not permit detailed mapping of the territory, especially in alpine areas [29]. Nevertheless, in recent years some institutional, academic centers and private enterprises have tried to overcome the spatial resolution issue by creating prototypal products at the national level by adopting Copernicus missions' EO Data [30,31]. New evidence is represented by the 10 m Land cover made by ESRI on a global scale with 10 classes, even if there are very strong limits and errors in the alpine area due to the absence of mountain training areas as well as ESA Global Land Cover product at 10 m, both based only on Sentinel-2 [32]. Another example linked to the Italian context is the prototypal LC of the whole Italian territory performed by ISPRA for the year 2018. This LC proposes a methodology with joint use of the optical multispectral and radar data of Sentinel 1 and Sentinel 2 [12]. However, following the choice of the input data adopted and the need to map an entire territory, it appears to have strong limits in the mountain area compared to the validation set; these areas are low and not adequate for mapping mountain territories in detail as suggested by [28,33,34].

The EO Data from the European Space Agency (ESA) and Sentinel-1 and Sentinel-2 satellites, as part of the Copernicus Earth monitoring program's Space component, have been exploited in this study [35]. On the one hand, Sentinel-1 interacts with elements through various signal polarizations (VV, VH) in proportion to their roughness and moisture content [36,37], and it uses synthetic aperture radar (SAR) imaging in the C band [38]. Its independence from weather conditions would make it a formidable tool in the mountains if it were not for the distortions that the radar signal is characterized by in the presence of geomorphologically complex surfaces; therefore, its use must be defined according to the type and location of the target area to map. On the other hand, the Sentinel-2 constellation has a 5-day temporal resolution and 13-band multispectral images ranging from visible to SWIR. Sentinel-2 data properties ensure good results in land cover monitoring [39,40] but shadows and clouds limit their application in mountain areas. Many studies have shown that combining Sentinel-1 and Sentinel-2 data overcomes the limitations of using single data products in land cover classification [41,42], but no one has explored limits and potentialities in mountain area land cover mapping [43]. Land cover detection (Joshi et al., 2016) can be improved by combining these two data sources, for example, for grassland [19],

urban regions [44], land cover changes [45], and hazard assessment [46–48]. Therefore, the development of an LC mapping continuous service according to the new EAGLE legend criteria is becoming of great interest to the public administration at the alpine level and beyond [49]. In fact, at the national level in Italy, ISPRA produces and updates the National Land Consumption Map as well as several regional land use and LC maps [30]. These regional products are frequently made with CLC and are not up to date [50,51]. The coarse geographical characteristics in the mountain areas do not allow a correct definition of the components of the territory, which are often below the minimum mapping unit indicated for the CLC but which together have a strong impact on the alpine territory and whose characterization may depend on access to forms of financing at the local level in the CAP and policies insurances [52]. In particular, many studies have adopted Sentinel-1 or Sentinel-2 to map land cover and other land cover components in some alpine areas, but nobody has studied the limits and potentialities offered by a coupled adoption of SAR and multispectral data in mountain areas [53–56].

Finally, in order to achieve the main goal of this work, a scalable worldwide approach capable of mapping EAGLE Land Cover in mountain areas with higher accuracies has been developed, overcoming remote sensing limitations and trying to exploit only the potentialities offered by radar and optical in complex geomorphological areas. In particular, an alpine suitable operative procedure to map with high spatial and temporal resolution and update the frequency of land cover for environmental planning and management following the European guidelines of EAGLE has been created. The product realized starting from the approach developed is compatible with the old Corine Land Cover and new rules in terms of the kind of classes that have to be adopted as well as creating a continuous service to help the alpine region to monitor a huge amount of component of the territory at a higher spatial resolution to help local, national, European and international planners [39,41] for the production of pixel-based land cover classification products [57].

As described above, the proposed approach was thought to overcome some of the limits that the classification of mountain areas generally introduces, mainly associated with local topography, weather conditions, shadows, and land cover class spatial distribution/fragmentation. This methodology has been developed in the Aosta Valley Autonomous region in NW Italy due to its higher degree of geomorphological complexity [6,9,58]. Because of these characteristics, RS processing and classification with high accuracies and great spatial detail of the degree of each LC component are very hard to perform, and, therefore, this region represents the perfect operational environment to perform the suggested method.

## 2. Materials

### 2.1. Study Area

The development of a possible approach to map LC according to EAGLE guidelines using optical and radar EO data in mountain areas has involved the Aosta Valley Autonomous Region in the NW of Italy. This territory has been chosen for the following reasons: (1) the highly complex aspect due to its morphological characteristics, which perfectly represents the typical mountainous conditions researched to develop a tentative robust alpine EAGLE classification approach; (2) the worst remote sensing operating environment due to the territory complexity (as described above) that make some techniques harder to be performed limiting the potential of EO data application. Therefore, to try to compensate for these limitations, it is necessary to adopt different workflows by identifying the most suitable to obtain the maximum results by combining at the end the layers obtained going beyond a single application of an ordinary machine learning algorithm. Here below, we report the study area involved in the development of the present workflow (see Figure 1).

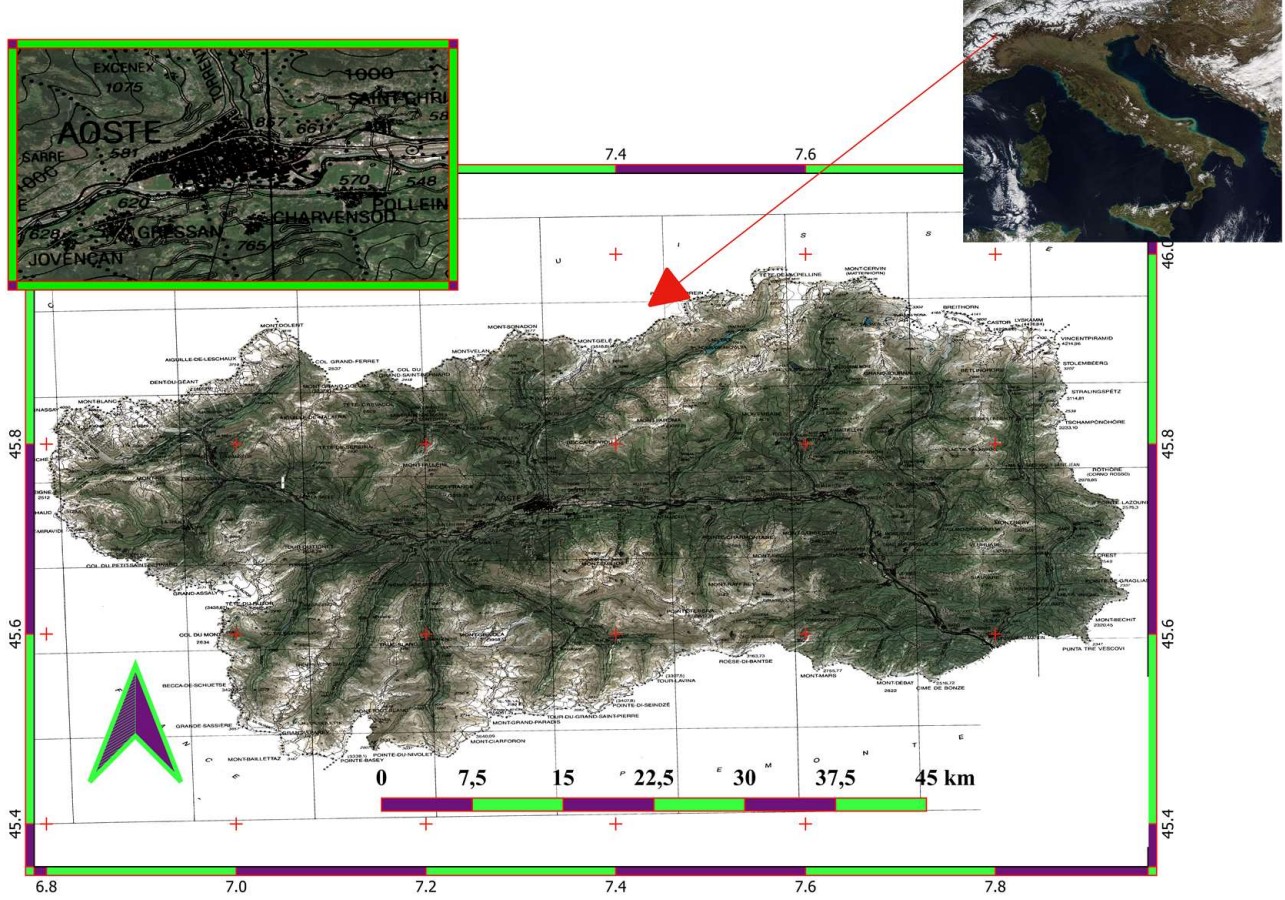

**Figure 1.** Study Area corresponding to the boundaries of the Aosta Valley Autonomous Region (NW Italy) (EPSG:23032).

It is worth noting that remote sensing limitations in this area, as well as in mountain areas worldwide, can be overcome reasonably by adopting a step-by-step hierarchical classification approach (as presented in this work) by detecting which algorithm seems to best map land cover components and exploit the best workflow per each class so as to optimize accuracy and reduce error. The question is not so much linked to forms of manual compensation of spectral or geometric signatures but to the careful use of satellite sensors on the basis of the little scientific literature present on this topic in the mountain area and according to the type of coverage and algorithm tests of machine learning as conducted in this study in order to define a scalable approach that allows a technology transfer.

### 2.2. Sentinel-1 SAR Data

*Copernicus Sentinel-1* mission is part of the European space program. The satellite acquires radar data with a temporal resolution of 5 days and a spatial resolution between 5 to 40 m depending on the acquisition mode. The radar data were obtained by the NASA Alaskan Satellite Facility (ASF) and processed in SNAP [59] and Google Earth Engine (GEE) [60]. The Sentinel-1 (hereinafter called S1) mission provides data from a dual-polarized C-band SAR (Synthetic Aperture Radar) instrument. Google Earth Engine provides only Sentinel-1 Ground Range Detected (GRD) collection, processed using Sentinel-1 Toolbox to generate a calibrated and ortho-correct product. Level-1 data can be processed into either Single Look Complex (SLC) and/or Ground Range Detected (GRD) products. The absence of Single Look Complex (SLC) data in GEE is due to the higher bit weight of this product and difficulty in the processing phase outside the SNAP environment. In particular, Level-1 SLC (IW) Interferometric Wide products (IW) were adopted [61,62]. IW swath mode is the main acquisition mode over land

and satisfies the majority of service requirements. It acquires data with a 250 km swath at 5 m by 20 m spatial resolution (single look) [34].

As mentioned before, for this work, SLC IW data were adopted by creating two separate datasets considering the same orbit, frame, and path of the scene in the study area. The two time series stacks ranging from all images ranging from 1 January 2020 to 31 December 2020 in ascending and descending mode, respectively. Those characteristics are reported in Table 1. As reported by [63], the main distortion in SAR data is the elevation displacement. The displacement increases with decreasing incidence angle. Therefore, the main SAR distortions in mountain areas are represented by: foreshortening, layover, and shadowing.

**Table 1.** SAR Stacks parameters criteria.

| Absolute Orbit Number | Polarization | Frame | Path | Flight Direction |
|:---:|:---:|:---:|:---:|:---:|
| 24,789 | VV + VH | 146 | 88 | ASCENDING |
| 24,417 | VV + VH | 441 | 66 | DESCENDING |

The ascending and descending were both processed in SNAP and then imported in GEE to create a mosaicked-median composite to reduce geometrics distortions in slopes where, normally, in a given acquisition mode occurs.

*2.3. Multispectral Optical Data*

2.3.1. Sentinel-2

*Sentinel-2* (hereinafter called S2) mission is part of the Copernicus European space program. The satellite acquires multispectral optical data with a spatial resolution between 10–20 m as a function of the band considered. The temporal resolution is 5 days being two twin satellites, S2A and S2B. The multispectral optical data were obtained and processed in Google Earth Engine (GEE) referring to the COPERNICUS/S2_SR collection. Sentinel-2 is a high-resolution, broad-spectrum, multispectral optical mission that supports Copernicus Land Monitoring studies, including monitoring of vegetation, soil, and water cover, as well as observation of inland waterways and coastal areas. Sentinel-2 L2 data are downloaded from Copernicus Scihub (the official distribution portal of the Earth Observation data in question). The images were pre-processed in sen2cor (official tool released by the European Space Agency—ESA). The EO data S2 pre-processed in sen2cor contains 12 spectral bands. The images are ortho-projected in WGS84 and are in-ground reflectance rescaled in dimensionless values from 0 to 10,000 starting from the DN from which they exist the radiances to calculate ground reflectance by removing the atmospheric contribution. There are also three QA bands for each scene, one of which (QA60) is a bitmask band with cloud mask information. In GEE, clouds can be removed as an alternative to using pixels in QA quality using COPERNICUS/S2_CLOUD_PROBABILITY. In this case, QA have been used, and bands up to 10 m GSD were bilinearly resampled at 10 m.

A yearly median composite imagery ranging from 1 May 2020 to 31 September 2020 without clouds and shadows has been realized. The S2 data have also been used in order to create yearly harmonized filtered NDVI and NDRE stacks with a 10-day step to map some vegetation classes, especially vineyards and orchards.

2.3.2. Planet EO Data

*Planetscope*, as part of the private space program Planet acquired by Google with its ultra-high spatial resolution microsatellites, is increasingly becoming a reference reality in remote sensing activities thanks to the fact that there is the possibility to have access to the data free of charge for Education and Research purposes (https://www.planet.com/markets/education-and-research/ last accessed on 6 November 2022). Starting from the daily data acquired by the Planetscopes, a composite imagery was generated for the same reference period as the S2 dataset used for an extra product in the validation phase and in the definition of the training sets in the photo-interpretation phase of some areas. The satellite acquires multispectral optical data on a daily basis in four bands

with a ground sample distance (GSD) at around 3 m with various levels of processing. In this case, georeferenced and atmospheric-calibrated products in Surface Reflectance were adopted. Considering that these data are not open-access and have a fee except for scientific purposes can be considered optional in the approach proposed in case of scaling to other areas. Certainly, they represent a useful tool to refine the map produced during a photo-interpretational phase.

### 2.4. GIS Products and Ground Data

*Digital Terrain Model (DTM)* of the Autonomous Region of Aosta Valley with a 10 m grid step acquired with flight lidar sensors in 2008 cropped and repositioned with perfect correspondence and overlapping between S1 and S2 pixels.

*Training set:* the polygons to train the classifier were defined by segmentation by objects (OBIA) followed by analysis of the spectral signatures and photo-interpretation of the image as well as from ground truth data polygon (GCP).

*Validation set:* polygons to validate the classification were obtained both through photo-interpretation and in the field, obtaining GCP. The validation was carried out in two phases: the first by calculating the confusion matrix by adopting the dataset obtained from S1 and S2 bands processing, and the second by assessing the manual classification. Finally, a Garmin 64S and the Lemon GPS smartphone application developed by the GeneGIS company were used to define the ground control points (GCP) bounded as polygons. The GNSS data were acquired in the Aosta Valley region in well-known classes. The boundaries were defined through perimeter detection or a-posteriori through photo-interpretation of Planet images. Here below, we report an overview (see Figure 2).

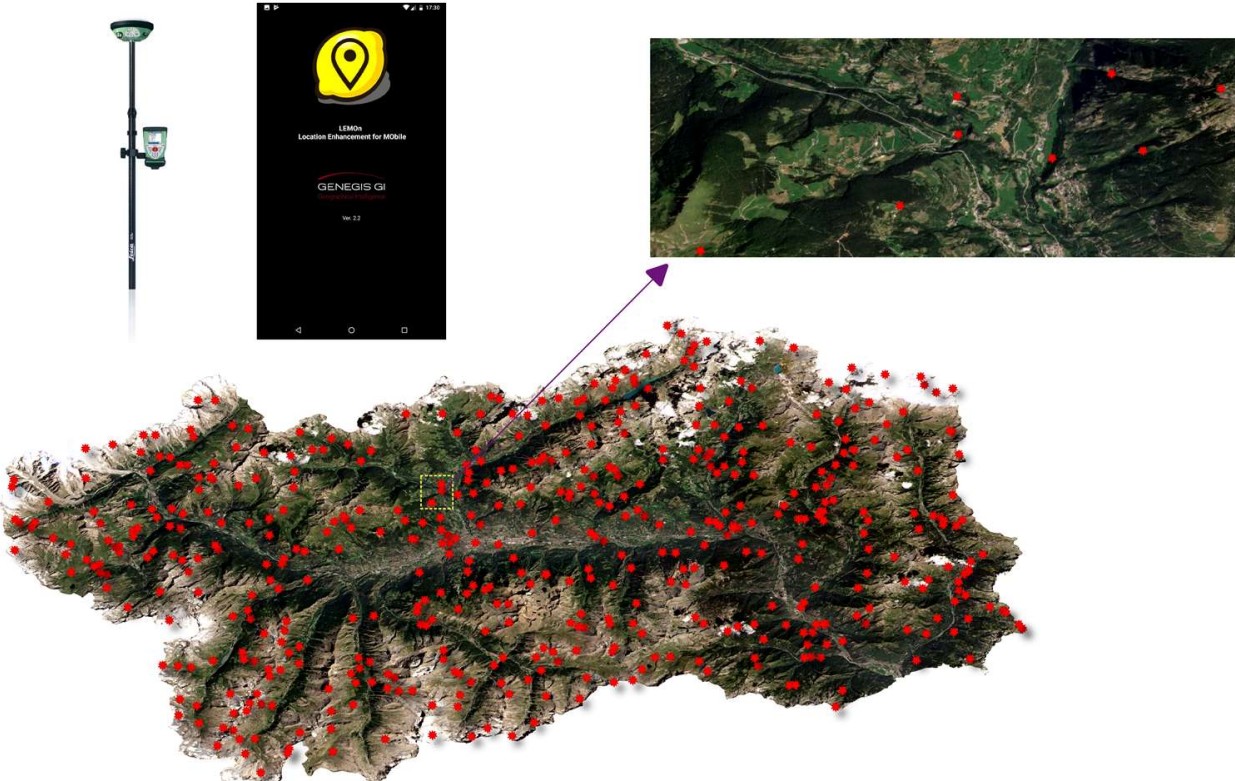

**Figure 2.** GNSS points detected and adopted a part as training set and another as validation set.

The collection of such data allowed us to populate both the training set and the validation. In particular, a random GCP selection was performed in SAGA GIS vers. 8.0.0 [64] with an allocation of 75% of the GCP to the training set and 25% of GCP to the validation set. The software adopted were: GEE [60], SNAP vers. 8.0.0 to obtain and calibrate the data during the pre-processing phase, Orfeo Toolbox vers 8.0.0 [65,66] SAGA GIS vers. 8.0.0 [64] to perform the classification during the processing phase and QGIS with GRASS and R v.3.0.1 [67–70] during the post-processing phase to prepare the final product.

## 3. Methods

### 3.1. Dataset Pre-Processing and Multi-Bands Stack Creation

The S1 and S2 data have been calibrated and processed in SNAP in GEE, respectively. S1 SAR data were used only to map urban and water component, respectively, in addition to optical data. The other classes were mapped only with optical remote sensing due to the fact that SAR distortions in mountain areas do not permit mapping at a higher accuracy of the land cover. Therefore, the data offered by optical remote sensing are the only ones in an alpine environment that are truly capable of offering consistent and reliable mappings despite being bound to atmospheric conditions but which, thanks to composite in land cover, it is possible to overcome. Calibration workflow to obtain GRD product was performed according to the [71] approach only for the water areas. Then, Normalized Difference Polarization Index (NDPI) and the Cross-Ratio (CR) were computed to analyze the water and humid areas. The urban areas were detected by interferometric analysis of coherence. Considering the GRD product, four S1 stacks were realized considering the ascending and descending modes (please see SAR Data in materials) for VV and VH bands adopted during the computation of NDPI and CR. These stacks of bands were finally clipped by using an aspect layer retrieved by the 10 m DTM VDA in those areas where SAR geometrical distortion normally affects portion of the imagery acquired in ascending or descending mode. Starting from the ancillary and metadata files, the angle of look, and the aspect layer ($\alpha$) have been considered during the clipping to exclude areas affected by strong distortions in both ascending and descending modes. Finally, the stacks have been mosaicked in order to fill the gaps created in each stack due to the removal of areas affected by strong distortions; in case of both distortions, we considered those portions with higher incidence angles according to [46,63]. This operation was performed in SAGA GIS. Then, the final stack obtained was uploaded in GEE to create a yearly SAR synthetic composite to compute NDPI and CR, as previously mentioned. The SAR composite was used to better map the water component, as previously mentioned.

### 3.2. Water SAR Mapping

In order to assess water areas components, the following SAR bands and indexes were adopted after a pre-processing phase explained in Table 2 and the creation of a composite to reduce SAR distortions.

**Table 2.** SAR Sentinel-1 GRD bands adopted in water mapping.

| | | MAIN INPUT DATASET S1 GRD |
|---|---|---|
| **ID** | **Bands/Index** | **Description** |
| 1 | "VV | Single co-polarization, vertical transmit/vertical receive |
| 2 | "VH" | Dual-band cross-polarization, vertical transmit/horizontal receive |
| 3 | "VV_STD" | Standard deviation Single co-polarization, vertical transmit/vertical receive |
| 4 | "VH_STD" | Standard deviation Dual-band cross-polarization, vertical transmit/horizontal receive |
| 5 | "NDPI" | Normalized Difference Polarization Index |
| 6 | "NDPI_STD" | Standard deviation Normalized Difference Polarization Index |
| 7 | "CR" | Cross ratio |
| 8 | "CR_STD" | Standard deviation Cross ratio |

NDPI and CR has been calculated as follow:
NDPI Normalized Difference Polarization Index [72]

$$\text{NDPI} = \frac{\text{VH} - \text{VV}}{\text{VH} + \text{VV}}$$

CR Cross-ratio [72]

$$\text{CR} = \frac{\text{VH}}{\text{VV}}$$

As demonstrated by [73], in complex morphological contexts, SAT approach seems to be more effective than the Otsu thresholding method. Therefore, these bands were included to map surface water areas by an automatic thresholding (SAT) approach [74]. The SAT approach consists of the following steps: (1) SAR data pre-processing to create a backscattering coefficient that is georeferenced with high-resolution LiDAR-derived DEM (in this case, the Aosta Valley DEM with 2 m step resampled at 10 m). (2) to compensate for elevation displacement, SAT and de-speckle filter were performed. (3) Conversion to dB (performed in SNAP vers.8.0.0). Indeed, the intensity of the radar signal reflected from the unit area of the corresponding point in the scene determined the pixel value of the SAR image, and the backscattering coefficient β0 was used in calibrating the surface object to convert the value from a digital number to reflectance. The radar cross-section of the target per unit area related to the local angle of incidence was the parameter β0. All SAR data were then converted from raw data to power units (decibel -dB). A de-speckle filter was used to remove salt and pepper noise while preserving edges and texture structures prior to data analysis due to the speckle effect produced by coherent radiation used in radar systems. A speckle Lee filter with a 5-pixel by 5-pixel window was employed, resulting in a unique valley-to-hill pattern in the histogram that could better distinguish between water and non-water surfaces. In addition, normalization between incident angles was performed. To identify a good threshold, we used a series of cubic polynomials to fit the histogram with some kind of moving step. This is because the cubic polynomial has the shape that best describes the histogram of the backscatter coefficients after de-speckle, and the inflection point is easier to solve than the higher-order polynomial.

Once the threshold had been applied, water non-water pixels were detected. The threshold value was determined by an iterative process to minimize intraclass variance while maximizing interclass variance. Finally, to refine the mapping of water areas, a supervised classification (Random Forest) was conducted in SNAP v.8.0.0 by adopting the main input pre-processed S1 GRD dataset (dividing the training set in water—not water).

*3.3. Urban SAR Mapping*

To better map the urban areas, as first step, pairs of S1 images were downloaded to map the urban areas and realize a layer. So, to perform interferometry with accurate repeatable coverage, we only take into account the images from the same satellite sensor in the correct acquisition mode (ascending or descending, see Table 1). In the last years in Aosta Valley, built-up area expansions have assumed a low rate, and, therefore, urban areas can be acquired within a single-year time frame in S1 images. Interferometry workflow has been performed according to [46,75]. Furthermore, interferometry was conducted only on those image pairs which have, within the year (2020), a perpendicular baseline of possibly more than 130 m and a temporal baseline lower than 10 days. Here we reported the pairs adopted available from ASF (see Table 3).

**Table 3.** SAR S1 images pairs adopted according to orbit in Table 2.

| S1 Pairs Ascending Orbit (Product No, Baseline, Temporal Distances in Days between the Two Acqusitions) | | | |
|---|---|---|---|
| S1A_IW_SLC__1SDV_20200430T172327 _20200430T172354_032360_03BEE8_2356 | S1B_IW_SLC__1SDV_20200506T172238 _20200506T172305_021464_028C15_773E | 136 m | 5d |
| S1B_IW_SLC__1SDV_20200530T172240 _20200530T172307_021814_029680_5539 | S1A_IW_SLC__1SDV_20200605T172329 _20200605T172356_032885_03CF21_34AB | 152 m | 7d |
| S1A_IW_SLC__1SDV_20200804T172333 _20200804T172400_033760_03E9BC_E6AD | S1B_IW_SLC__1SDV_20200810T172255 _20200810T172322_022864_02B66E_1179 | 152 m | 6d |
| S1A_IW_SLC__1SDV_20200828T172334 _20200828T172401_034110_03F5FE_8B79 | S1B_IW_SLC__1SDV_20200903T172253 _20200903T172320_023214_02C15A_3F08 | 162 m | 6d |
| S1B_IW_SLC__1SDV_20200903T172253 _20200903T172320_023214_02C15A_3F08 | S1A_IW_SLC__1SDV_20200909T172335 _20200909T172402_034285_03FC20_A288 | 159 m | 6d |
| S1B_IW_SLC__1SDV_20201009T172254 _20201009T172321_023739_02D1C8_57D8 | S1A_IW_SLC__1SDV_20201015T172336 _20201015T172402_034810_040E9B_A403 | 134 m | 6d |
| S1B_IW_SLC__1SDV_20201114T172240 _20201114T172307_024264_02E22F_E4D7 | S1A_IW_SLC__1SDV_20201120T172335 _20201120T172402_035335_0420C3_E828 | 144 m | 7d |
| S1 Pairs Ascending orbit | | | |
| S1A_IW_SLC__1SDV_20200112T053523 _20200112T053550_030763_03871C_D73E | S1B_IW_SLC__1SDV_20200118T053455 _20200118T053522_019867_02592E_ADC0 | 165 m | 5d |
| S1B_IW_SLC__1SDV_20200211T053455 _20200211T053522_020217_026479_497E | S1A_IW_SLC__1SDV_20200217T053522 _20200217T053548_031288_03996E_2722 | 155 m | 7d |
| S1A_IW_SLC__1SDV_20200324T053522 _20200324T053549_031813_03ABB5_4955 | S1B_IW_SLC__1SDV_20200330T053455 _20200330T053522_020917_027ABA_DC4C | 129 m | 5d |
| S1B_IW_SLC__1SDV_20200505T053456 _20200505T053523_021442_028B5C_A52F | S1A_IW_SLC__1SDV_20200511T053523 _20200511T053550_032513_03C3F4_2251 | 138 m | 7d |
| S1B_IW_SLC__1SDV_20200118T053455 _20200118T053522_019867_02592E_ADC0 | S1A_IW_SLC__1SDV_20200124T053522 _20200124T053549_030938_038D40_8123 | 147 m | 7d |

ESA guidelines [76,77] were used by introducing a variation in the type of classification; in this case, not Maximum Likelihood but Random forest and batch processing was created thanks to a routine to involve all pairs selected. The processing steps to correctly calibrate SAR images in order to map and discriminate urban and not urban areas have been reported here below. It is worth noting that these steps have been realized in the ESA SNAP v.8.0.0 toolbox (see more detail in Figure 3).

In the workflow, we selected just those bursts that covered our study area (the Aosta Valley Autonomous Region) from the original product. In addition, the coherence estimation was performed by using a default window range of 20 pixels. Finally, a Range-Doppler terrain correction was performed, which entails using the 10 m Aosta Valley DTM implemented in SNAP repository, selecting ED50-UTM 32 N (EPSG: 23032). The output coherence product consists of two bands per each polarization. It is worth noting that coherence between two SAR images expresses the similarity of the radar reflection between them. Any changes in the complex reflectivity function of the scene are manifested as a decorrelation in the phase of the appropriate pixels between the two images. Finally, from this output, a supervised classification was performed. Since we were interested in mapping urban and non-urban areas, we aggregated all non-urban land cover types into the same class (such as glaciers, lawn pastures, needle forests, and so on). A Random Forests classifier was performed in SNAP, and we specified the maximum number of decision trees in the RF classifier, which we set at 500 as the optimal value to achieve noise cancellation and smooth response [44].

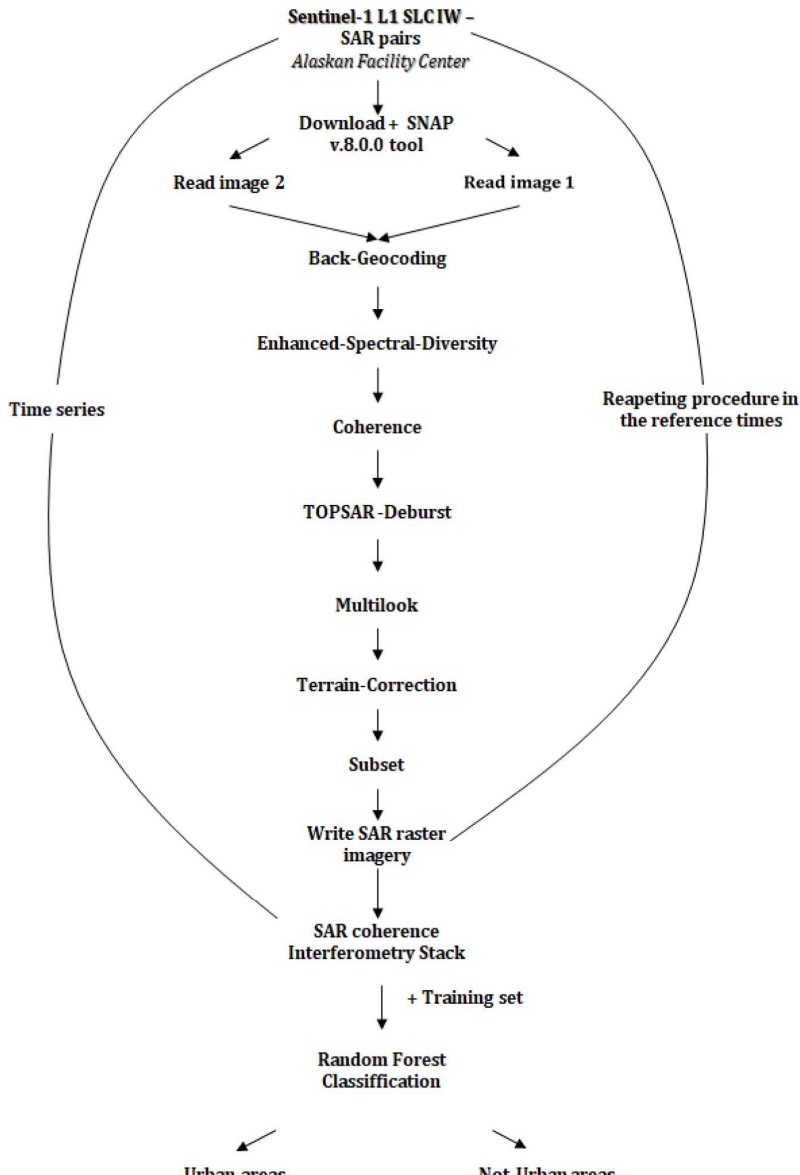

**Figure 3.** Interferometry in ESA SNAP v.8.0.0 and classification VH and VV were adopted.

The classification images produced from S1 imagery consist of discrete raster, with all pixels classified into either "urban" or "non-urban" values (with values 1 and 0, respectively) and water or "non-water".

Urban and water masks were adopted in the following phase using optical data to better refine these classes, which are unique and less affected by SAR distortions in mountain areas. The refining between SAR and optical data were performed manually.

*3.4. Multispectral Sentinel-2 Mapping*

The S2 data were retrieved by GEE collection COPERNICUS/S2_SR that was already calibrated in surface reflectance. A self-made algorithm performed in GEE was adopted to create a median composite. The S2 composite stack includes bands, spectral indices, and standard deviations. These input parameters have been reported in Table 4. The S2 stack with DTM and slope was adopted as input data during the classification, while the S1 output layers only to better refine urban and water classes. Each composite image was generated starting from the EO Data available every 5 days for the period from 1 May 2020

to 30 September 2020 (t), i.e., the summer weather season, in order to correctly map the glacial surface of the territory falling within the ablation period and observes the vegetation during the active phenological period. The composite images generated consist of the median value for each pixel in the reference period t. For S2, we considered all the images that satisfy the condition in which each pixel has cloud cover equal to zero (the clouds and shadows have been suitably masked, and the pixel, if cloudy, has not been considered in the definition of the median value of the reflectance of each band). After pre-processing S1 and S2 in GEE through a self-made algorithm, the bands and the indices shown in Table 4 were generated, and the standard deviation was calculated for each band as follows:

**Table 4.** Composition of the input multispectral datasets in the mountain areas analyzed.

| ID | Bands/Index | Description |
|---|---|---|
| 1 | "B2" | Blue |
| 2 | "B3" | Green |
| 3 | "B4" | Red |
| 4 | "B5" | Vegetation Red Edge 1 |
| 5 | "B6" | Vegetation Red Edge 2 |
| 6 | "B7" | Vegetation Red Edge 3 |
| 7 | "B8" | NIR |
| 8 | "B8A" | Vegetation Red Edge 4 |
| 9 | "B11" | SWIR 1 |
| 10 | "B12" | SWIR 2 |
| 11 | "B2_STD" | Standard deviation Blue |
| 12 | "B3_STD" | Standard deviation Green |
| 13 | "B4_STD" | Standard deviation Red |
| 14 | "B5_STD" | Standard deviation Red Edge 1 |
| 15 | "B6_STD" | Standard deviation Red Edge 2 |
| 16 | "B7_STD" | Standard deviation Red Edge 3 |
| 17 | "B8_STD" | Standard deviation NIR |
| 18 | "B8A_STD" | Standard deviation Red Edge 4 |
| 19 | "B11_STD" | Standard deviation SWIR 1 |
| 20 | "B12_STD" | Standard deviation SWIR 2 |
| 21 | "NDVI" | Normalized Difference Vegetation Index |
| 22 | "NDVI_STD" | Standard deviation Normalized Difference Vegetation Index |
| 23 | "BSI" | Bare Soil Index |
| 24 | "BSI_STD" | Standard deviation Bare Soil Index |
| 25 | "NDWI" | Normalized Difference Water Index |
| 26 | "NDWI_STD" | Standard deviation Normalized Difference Water Index |
| 27 | "NDSI" | Normalized Difference Snow Index |
| 28 | "NDSI_STD" | Standard deviation Normalized Difference Snow Index |
| 29 | "TCB" | Tasselled Cap Brightness |
| 30 | "TCB_STD" | Standard deviation Tasselled Cap Brightness |
| 31 | "TCG" | Tasselled Cap Greenness |
| 32 | "TCG_STD" | Standard deviation Tasselled Cap Greenness |
| 33 | "TCW" | Tasselled Cap Wetness |
| 34 | "TCW_STD" | Standard deviation Tasselled Cap Wetness |
| 43 | DTM | Digital Terrain Model 10 m |
| 44 | Slope | Terrain Slope |
| 45 | Aspect | Terrain aspect |

The spectral indexes reported in Table 5 have been calculated as follow in Table 5:

Since orchards and vineyards as well as permanent crops were particularly complex to discriminate (hereinafter called AGR) after performing a single classification due to the fact that a single multispectral composite input dataset did not permit considering the whole phenological active season, a hierarchical classification approach was then implemented. It has foreseen first classification (considering S2 Main Input dataset) with all the classes according to the new EAGLE Land Cover Legend (17 classes in this case) and a subsequent one with only these classes, masking all the remaining ones. In the



end, the two classifications were subjected to a mosaicking process by applying a first overlapping for orchards and vineyards. Then, the doubtful areas were corrected manually by photo-interpretation of a composite Planetscope imagery.

**Table 5.** Multispectral indexes.

| Spectral Index | Formula |
| --- | --- |
| NDVI<br>Normalized Difference Vegetation Index [78–82] | $\text{NDVI} = \frac{\text{NIR}-\text{RED}}{\text{NIR}+\text{RED}}$ |
| BSI<br>Bare Soil Idex [83] | $\text{BSI} = \frac{(\text{SWIR 1}+\text{RED})-(NIR+BLUE)}{(\text{SWIR 1}+\text{RED})+(NIR+BLUE)}$ |
| NDWI<br>Normalized Difference Water Index [84,85] | $\text{NDWI} = \frac{\text{NIR}-\text{SWIR 1}}{\text{NIR}+\text{SWIR 1}}$ |
| NDSI<br>Normalized Difference Snow Index [86–89] | $\text{NDSI} = \frac{\text{NIR}-\text{SWIR 1}}{\text{NIR}+\text{SWIR 1}}$ |
| TCB<br>(Tasselled Cap Brightness) [90–94] | $(\text{BLUE} \times 0.3037) + (\text{GREEN} \times 0.2793 + (\text{RED} \times 0.4743)$<br>$+ (\text{NIR} \times 0.5585) + (\text{SWIR1} \times 0.5082) + (\text{SWIR2} \times 0.1863$ |
| TCG<br>(Tasselled Cap Greenness) [90–94] | $(\text{BLUE} \times -0.2848) + (\text{GREEN} \times -0.243) + (\text{RED} \times -0.5436)$<br>$+ (\text{NIR} \times 0.7243) + (\text{SWIR1} \times -0.0840) + (\text{SWIR2} \times -0.1800)$ |
| TCW<br>(Tasselled Cap Wetness) [90–94] | $((\text{BLUE} \times 0.1509) + (\text{GREEN} \times 0.1973) + (\text{RED} \times 0.3279)$<br>$+ (\text{NIR} \times 0.3406) + (\text{SWIR1} \times -0.7112) + (\text{SWIR2} \times -0.4572))$ |

In this regard, to better map AGR, S2 yearly images (2020) were adopted, performing a supervised Minimum Distance classification (MDC) starting with the following inputs normalized:

- Yearly cloud-shadow masked NDVI stack filtered (Savitzky–Golay) [95–97] and regularized at 10 days times-steps [98] on GEE.
- Annual stack of the NDRE index (Normalized Difference Red Edge Index for Agriculture) following the same procedure of NDVI stack [99]

$$\text{NDRE} = \frac{\text{NIR} - \text{RE}}{\text{NIR} + \text{RE}}$$

- NDVI composite Entropy [11,58]

$$\text{H}_{\text{NDVI}} = - \sum_{i=0}^{N-1} \sum_{j=0}^{N-1} \text{NDVI}_{i,j} \log(\text{NDVI}_{i,j})$$

where $\text{NDVI}_{i,j}$ is the NDVI value at the i-th row and j-th column in the local square window measuring N pixels. For this study, a kernel window size of $10 \times 10$ pixels was adopted.

- Rao's Q Diversity Index on S2 NDVI composite [100]. Rao's Q is calculated using half the squared Euclidean distance, therefore, the resulting index is [101]:

$$\text{Q} = \sum \sum \text{d}_{ij} \times \text{p}_i \times \text{p}_j$$

where $\text{p}_i$ and $\text{p}_j$ are, respectively, the proportion of area of each category per rows and columns in the pairwise distance $\text{d}_{ij}$.

- Pattern analysis on S2 NDVI composite. The following pattern was computed: Dominance, Diversity, Relative richness, and Fragmentation. The settings parameters were: maximum number of classes: 17; kernel type: circle; radius 2.

To better discriminate these classes, the above-mentioned procedure was adopted because a simple classification with only median spectral sign and median indexes is not able to carefully detect agronomic cultivation, as demonstrated by [10]. NDVI and NDRE temporal stacks were considered, including textural patterns, in order to carefully map these classes considering that they are very complex to detect, especially in mountain areas. This procedure has permitted a rise in the user accuracy for both classes of 27%, reaching per each class mapped a user accuracy of more than 90%.

Then, the KMC were mosaicked using as first overlap onto the MDC retrieved to refine only AGR classes. As the last step, a simple filter was performed using a radius greater than 20 m. Furthermore, the classification was manually refined by a photo-interpretation process, also using Planetscope imagery to improve the accuracy of the manufacturer and the user and overall reduce errors. Finally, a confusion matrix was computed from the validation set. The parameters reported in the confusion matrix [102] were: Overall accuracy, Errors of Commission and Omission, User and Producer accuracy, Sum users and sum producers, and unclassified pixels (in this case, each pixel was classified).

Finally, the kappa coefficient measures the agreement between classification and truth values. A kappa value of 1 represents perfect agreement, while a value of 0 represents no agreement. The kappa coefficient is computed as follows:

$$k = \frac{N \sum_{i=1}^{n} m_{i,j} - \sum_{i=1}^{n} (G_j C_j)}{N^2 - \sum_{i=1}^{n} (G_j C_j)}$$

where:

$_i$: is the class number

N: is the total number of classified values compared to truth values

$m_{i,i}$: is the n° values belonging to the truth class $_i$ that have been classified as class $_i$

$C_i$: is the total number of predicted values belonging to class $_i$

$G_i$: is the total number of truth values belonging to class $_i$

Despite a semi-automatic workflow, as previously said, some manual photo-interpretation refining was performed involving urban and rock classes, as well as lawn pastures and alpine grasslands, respectively.

*3.5. Definition of Optimal Number of Area of Interest (AOI) Required for Each Class of the Training Set*

After creating the initial input dataset, a K-means unsupervised classification with 17 classes was performed in order to better know the spatial extent distribution of each class with the aim of defining the optimal number of training areas for each class of the training set. It is worth noting that there must be enough training pixels for each spectral class to allow for reasonable estimates of the elements of the conditional mean vector and of the class covariance matrix. According to [33,34], for the N-dimensional multivariance space, the covariance matrix is symmetric of size N × N. Therefore, it has 1/2 N (N + 1) distinct elements that must be estimated from the training data. In order to avoid the matrix being singular, at least N(N + 1) independent samples are needed. Fortunately, each N-dimensional pixel vector actually contains N samples (one of each waveband); therefore, the minimum number of independent training pixels required is (N + 1). Because it is difficult to guarantee the independence of pixels, it is common to choose more than this minimum number. Ref. [103] recommends, as a practical minimum, that 10 N training pixels per spectral class be used, with as many as 100 N per class if possible. Therefore, for this classification, considering only the spectral bands and relative indexes without the standard deviation, a minimal number of 250 polygons has been computed (including a minimum of 5 pixels) for each class.

*3.6. Stack Segmentation and AOI Definitions*

The Object-Based-Segmentation (OBS) approach was performed using the mean shift algorithm available on the Orfeo Toolbox software v.8.0.0 [104,105]. OBC algorithms

aimed at minimizing the spectral heterogeneity of polygons by comparing relatively the spectral properties of neighboring pixels. The resulting segmentation vector layer (SVL) was generated according to a previously defined minimum mapping unit of 300 m$^2$. In particular, the segmentation was performed with reference to the S2 bands having a Ground Sample Distance (GSD) equal to 10 m, i.e., B2, B3, B4, and B8 natively. Starting from the native reflectance values, the images were segmented into segments having an internally homogeneous spectral response. The segments were then vectorized to generate the corresponding vector layer. During segmentation, the required parameters were set to the values shown in Table 6. SEG was then used to explore internal features other than spectral signatures, such as recurrent radiometric patterns (texture) and shape. Some of these polygons were then randomly extracted, and others were created by analyzing the signatures of the entire stack to define part of the training areas. This procedure has permitted defining some training areas per each class.

**Table 6.** Segmentation parameters.

| Segmentation Parameter | Settings |
|---|---|
| Spatial radius | 3 pixels |
| Range radius | 100 DN |
| Mode convergence threshold | 0.1 |
| Maximum numerous of iterations | 200 |
| Minimum region size | 3 pixels |

*3.7. Regions of Interests Distributions*

ROI was defined mostly on the field and partially by applying both segmentation and a spectral signature-photo interpretation phase. The image below depicts the distribution of the ROIs in the study area. Each ROI per class has a number of polygon upper to 250. An overall of 4300 ROIs were defined, and 75% of them were adopted as training set the 25% as validation set (see Figure 4).

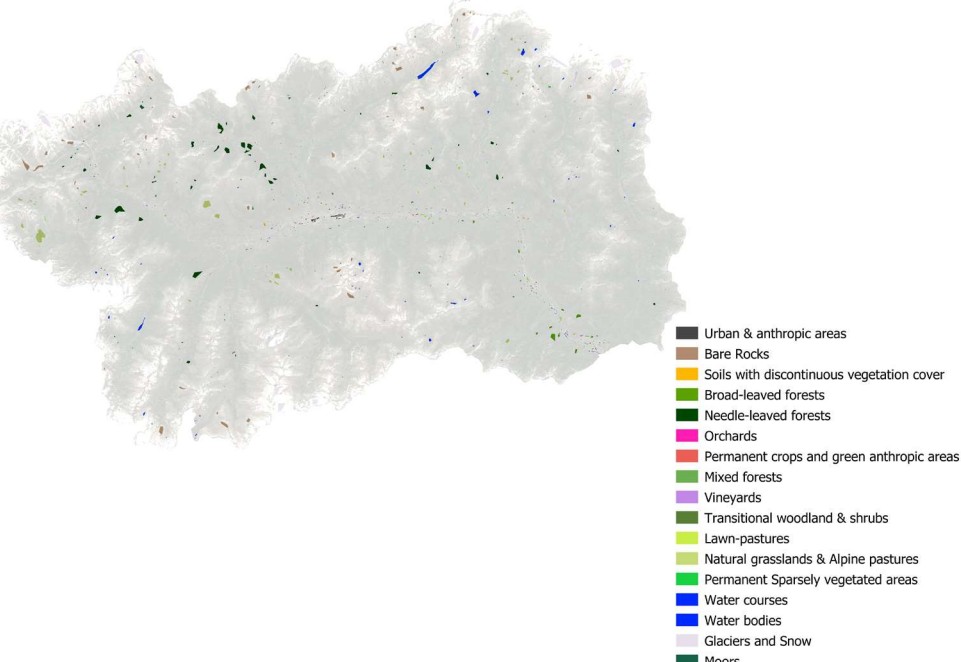

**Figure 4.** An overlook of ROIs distribution for both training and validation sets. Most of ROIs per classes have no more then 4 pixels therefore they cannot be appreciated in this map to have a general overlook please see Figure 2.

*3.8. Classification and Confusion Matrix*

Finally, starting from the input dataset and the training set, several supervised classifications were performed in SAGA GIS vers. 8.0.0 by adopting different machine learning algorithms, and the relative confusion matrix was computed in order to define the most suitable in complex morphology, such as mountainous Aosta Valley region, by evaluating the best overall accuracy and therefore minimizing the errors. Given the characteristics of the input dataset and the alpine territory analyzed, the best performing algorithm was the K-Nearest Neighbors Classification-KMC and the Minimum Distance with pre-segmentation (SNIC) by applying a Distance threshold of 50. The k-nearest neighbors (k-NN) is an algorithm used in pattern recognition for the classification of objects based on the characteristics of the objects close to the one considered. It is a non-parametric classification method. In both cases, the input is the closest k training examples in the feature space. The output depends on whether k-NN is used for classification or regression. In the k-NN classification, the output is a membership in a class. An object is classified by a plurality vote of its neighbors, with the object assigned to the most common class among its k closest neighbors (k is a positive, typically small, integer). If k = 1, the object is simply assigned to the class of that single closest neighbor. In the k-NN regression, the output is the property value for the object. This value is the average of the closest neighboring k values. On the other hand, the minimum distance classifier is used to classify unknown image data into classes, which minimizes the distance between the image data and the class in multi-feature space. The distance is defined as an index of similarity so that the minimum distance is identical to the maximum similarity. Therefore, the minimum distance technique uses the mean vectors of each endmember and calculates the Euclidean distance from each unknown pixel to the mean vector for each class. All pixels are classified to the nearest class unless a standard deviation or distance threshold is specified, in which case some pixels may be unclassified if they do not meet the selected criteria.

In order to ensure the scalability of the methodology proposed in the mountain area, we have tried to condense everything in the following workflow (see Figure 5):

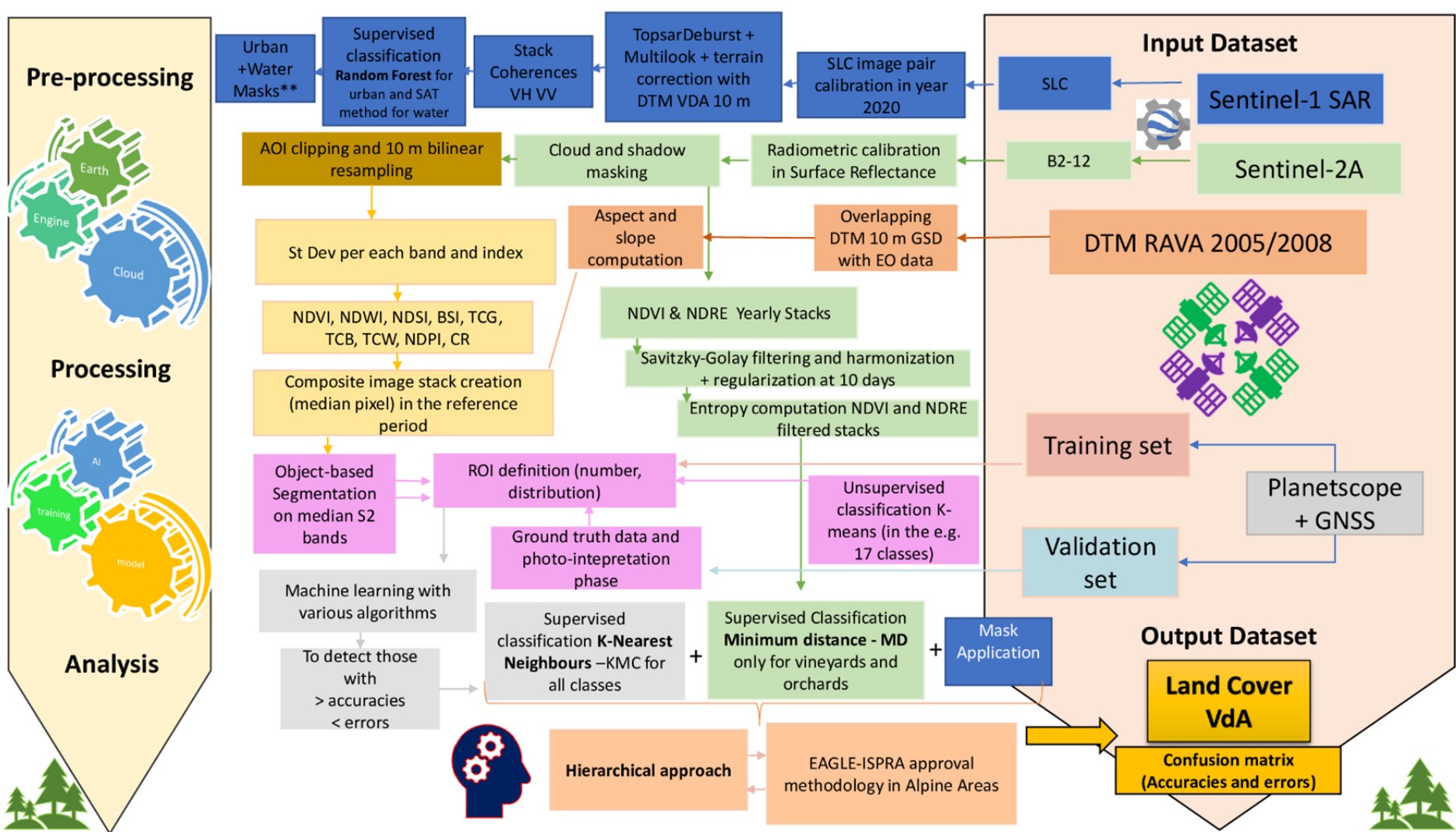

**Figure 5.** Workflow methodology developed to map EAGLE Land Cover in Alpine Areas.

## 4. Results

The supervised classifications were performed by adopting the following algorithms [16,106]: Supervised Maximum Likelihood, Minimum Distance with pre-segmentation (SNIC) [19,107,108], the Random Forest (OpenCV), Support Vector Machine (OpenCV), Artificial Neural Network—ANN (neural networks)—[108,109], and K-Nearest Neighbors Classification (OpenCV) [64]. Classifications performed were then vectorized. At the same time, the confusion matrix was calculated for each of the techniques adopted to identify the goodness of the product realized. The best one was chosen by observing the results offered by K-coefficient and the overall accuracy. In fact, the main aim was to find the optimal algorithm to minimize the errors and therefore maximize the accuracy. The two best were: K-Nearest Neighbors Classification (OpenCV) and the Minimum Distance with pre-segmentation (SNIC) with a distance threshold of 50. Here below, it was reported the overall accuracies and the K coefficient obtained by performing a confusion matrix for each classification realized (see Table 7).

**Table 7.** Classes mapped.

| EAGLE Land Cover Classes | Broad-leaved forests |
| --- | --- |
| Bare Rocks | Needle-leaved forests |
| Permanent crops and green anthropic areas | Mixed Forests |
| Soils with discontinuous vegetation cover | Lawn-pastures |
| Permanent sparsely vegetated areas | Natural grassland and Alpine pastures |
| Transitional woodland and shrubs | Orchards |
| Glaciers and snow | Vineyards |
| Moors | Water bodies |
| Urban and anthropic areas | Water courses |

The EAGLE classes mapped in mountain areas are the following in Table 7, and they are carefully described in Appendix A:

Concerning the quality reached in mapping a complex geomorphological area, here below a snapshot of some areas in the Aosta Valley region with a representation scale of 1:10,000 has been provided (the images in the upper part and 1:40,000 in the Aosta municipality (please see Figure 6).

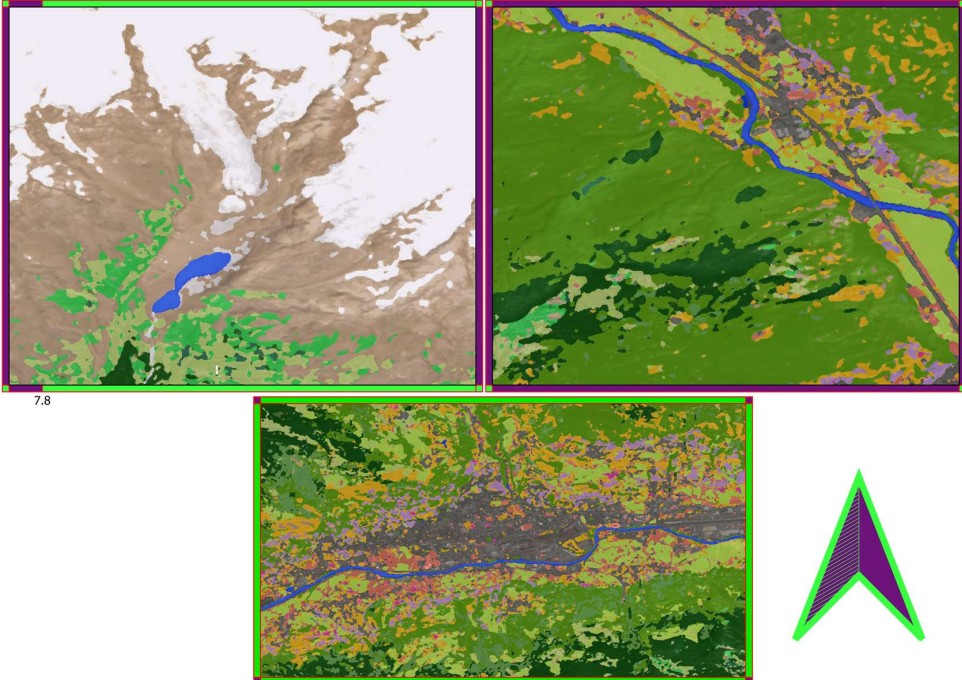

**Figure 6.** Samples images of classified areas with a high satellite resolution images on the background.

In order to better map urban and water areas components, S1 masks obtained were overlapped with urban and water surfaces mapped with S2 data. Then, manual refining was performed. In order to assess the classification land cover quality, several supervised classifications were realized, as reported in Table 8.

**Table 8.** Accuracies obtained from different machine learning algorithms to perform supervised classifications.

| Machine Learning Supervised Classification Algorithm | Overall Accuracy | K-Coefficient |
|---|---|---|
| K-Nearest Neighbors Classification (OpenCV) | 93% | 0.93 |
| Minimum Distance with pre-segmentation (SNIC) | 92% | 0.92 |
| Artificial Neural Network—ANN (neural networks) | 84% | 0.84 |
| Random Forest (OpenCV) | 88% | 0.88 |
| Support Vector Machine (OpenCV) | 90% | 0.90 |
| Supervised Maximum Likelihood | 85% | 0.85 |

Each classification was carried out by normalizing the dataset due to the diversity of the input variables to make them homogeneous. The parameters adopted in the K-Nearest Neighbors Classification-KMC (OpenCV) were the number of neighbors equal to 8 and a training method named: classification, and finally, the type of Brute Force algorithm [110]. K-Nearest Neighbors Classification (OpenCV) and Minimum Distance with pre-segmentation (SNIC) were adopted as algorithms in the present work. The workflow followed is reported in Figure 6. Here below, the improvement obtained by performing the two classifications has been reported, not only a single one, as described in the previous section. In both cases, no photo-interpretation refining was performed in this phase. In the end, accuracies were computed (see Table 9).

**Table 9.** Comparison of accuracies.

| Machine Learning Supervised Classification Algorithm | Overall Accuracy | K-Coefficient |
|---|---|---|
| K-Nearest Neighbors Classification (OpenCV) | 93% | 0.93 |
| K-Nearest Neighbors Classification (OpenCV + Minimum Distance) | 96% | 0.96 |

Finally, manual refining was performed using Planetscope imagery as described in the above section, then the final confusion matrix (CM) was computed, and the map was generated, as reported in Figures 7 and 8.

| CONFUSION MATRIX | CLASS | Urban & anthropic areas (11) | Permanent snow and ice (32) | Moors & heathlands (41) | Bare Rocks (121) | Sparsely vegetated areas (222) | Water courses (311) | Water surfaces (312) | Soils with reduced vegetation cover (1221) | Broad-leaved forests (2111) | Needle-leaved forests (2112) | Mixed forests (2114) | Orchards (21131) | Permanent crops and green antropic areas (21133) | Vineyards (21211) | Shrubland and transitional woods (21221) | Lawn-pastures (22111) | Natural grasslands & alpine pastures (22112) | Sum User | Accuracy User | Errors of Omission |
|---|---|---|---|---|---|---|---|---|---|---|---|---|---|---|---|---|---|---|---|---|---|
| Urban & anthropic areas | 11 | 1102 | 0 | 0 | 3 | 0 | 2 | 0 | 1 | 0 | 0 | 0 | 0 | 0 | 1 | 0 | 0 | 4 | 1113 | 99.01 | 0.010 |
| Permanent snow and ice | 32 | 0 | 23866 | 0 | 30 | 0 | 0 | 0 | 0 | 0 | 0 | 0 | 0 | 0 | 0 | 0 | 0 | 0 | 23896 | 99.87 | 0.001 |
| Moors & heathlands | 41 | 0 | 0 | 959 | 0 | 0 | 0 | 0 | 0 | 0 | 12 | 0 | 0 | 0 | 0 | 8 | 0 | 0 | 979 | 97.96 | 0.020 |
| Bare Rocks | 121 | 0 | 49 | 0 | 9031 | 2 | 0 | 0 | 0 | 0 | 0 | 0 | 0 | 0 | 0 | 1 | 0 | 1 | 9084 | 99.42 | 0.006 |
| Sparsely vegetated areas | 222 | 0 | 0 | 0 | 64 | 1057 | 0 | 0 | 0 | 0 | 0 | 0 | 0 | 0 | 0 | 4 | 0 | 37 | 1162 | 90.96 | 0.090 |
| Water courses | 311 | 33 | 0 | 0 | 0 | 0 | 1296 | 0 | 0 | 0 | 0 | 2 | 0 | 0 | 0 | 0 | 0 | 0 | 1331 | 97.37 | 0.026 |
| Water surfaces | 312 | 0 | 0 | 0 | 0 | 0 | 59 | 1319 | 0 | 0 | 0 | 0 | 0 | 0 | 0 | 0 | 0 | 0 | 1378 | 95.72 | 0.043 |
| Soils with reduced vegetation cover | 1221 | 2 | 0 | 0 | 0 | 0 | 0 | 0 | 615 | 23 | 0 | 0 | 1 | 1 | 11 | 0 | 1 | 0 | 654 | 94.04 | 0.060 |
| Broad-leaved forests | 2111 | 0 | 0 | 0 | 0 | 0 | 0 | 0 | 0 | 2285 | 0 | 34 | 11 | 5 | 1 | 2 | 0 | 0 | 2338 | 97.73 | 0.023 |
| Needle-leaved forests | 2112 | 0 | 0 | 0 | 180 | 0 | 0 | 0 | 0 | 14 | 24159 | 4 | 0 | 0 | 0 | 5 | 0 | 24 | 24386 | 99.07 | 0.009 |
| Mixed forests | 2114 | 0 | 0 | 0 | 0 | 0 | 0 | 0 | 0 | 0 | 0 | 982 | 0 | 0 | 0 | 0 | 0 | 0 | 982 | 100.00 | 0.000 |
| Orchards | 21131 | 0 | 0 | 0 | 0 | 0 | 0 | 0 | 0 | 0 | 0 | 0 | 238 | 0 | 0 | 0 | 0 | 0 | 238 | 100.00 | 0.000 |
| Permanent crops and green antropic areas | 21133 | 19 | 0 | 0 | 0 | 0 | 0 | 0 | 8 | 0 | 0 | 0 | 5 | 1717 | 1 | 0 | 6 | 0 | 1756 | 97.78 | 0.022 |
| Vineyards | 21211 | 0 | 0 | 0 | 0 | 0 | 0 | 0 | 22 | 0 | 0 | 0 | 0 | 0 | 3128 | 0 | 0 | 0 | 3150 | 99.30 | 0.007 |
| Shrubland and transitional woods | 21221 | 0 | 0 | 0 | 0 | 0 | 0 | 0 | 0 | 0 | 0 | 0 | 0 | 0 | 0 | 5483 | 0 | 0 | 5483 | 100.00 | 0.000 |
| Lawn-pastures | 22111 | 6 | 0 | 0 | 0 | 0 | 0 | 0 | 0 | 0 | 0 | 0 | 0 | 26 | 0 | 0 | 13633 | 32 | 13697 | 99.53 | 0.005 |
| Natural grasslands & alpine pastures | 22112 | 0 | 0 | 6 | 196 | 7 | 0 | 0 | 0 | 4 | 100 | 0 | 0 | 21 | 0 | 3 | 223 | 18132 | 18692 | 97.00 | 0.030 |
| **Sum Producer** | | 1162 | 23915 | 965 | 9504 | 1066 | 1357 | 1319 | 646 | 2326 | 24271 | 1022 | 255 | 1770 | 3142 | 5506 | 13863 | 18230 | 110319 | | |
| **Unclassified** | | 0 | 0 | 0 | 0 | 0 | 0 | 0 | 0 | 0 | 0 | 0 | 0 | 0 | 0 | 0 | 0 | 0 | **Overall Accuracy** | | **K Coefficient** |
| **Accuracy Producer** | | 94.84 | 99.80 | 99.38 | 95.02 | 99.16 | 95.50 | 100.00 | 95.20 | 98.24 | 99.54 | 96.09 | 93.33 | 97.01 | 99.55 | 99.58 | 98.34 | 99.46 | **0.98** | | **0.98** |
| **Errors of Commission** | | 0.052 | 0.002 | 0.006 | 0.050 | 0.008 | 0.045 | 0.000 | 0.048 | 0.018 | 0.005 | 0.039 | 0.067 | 0.030 | 0.004 | 0.004 | 0.017 | 0.005 | | | |

**Figure 7.** Final Confusion matrix.

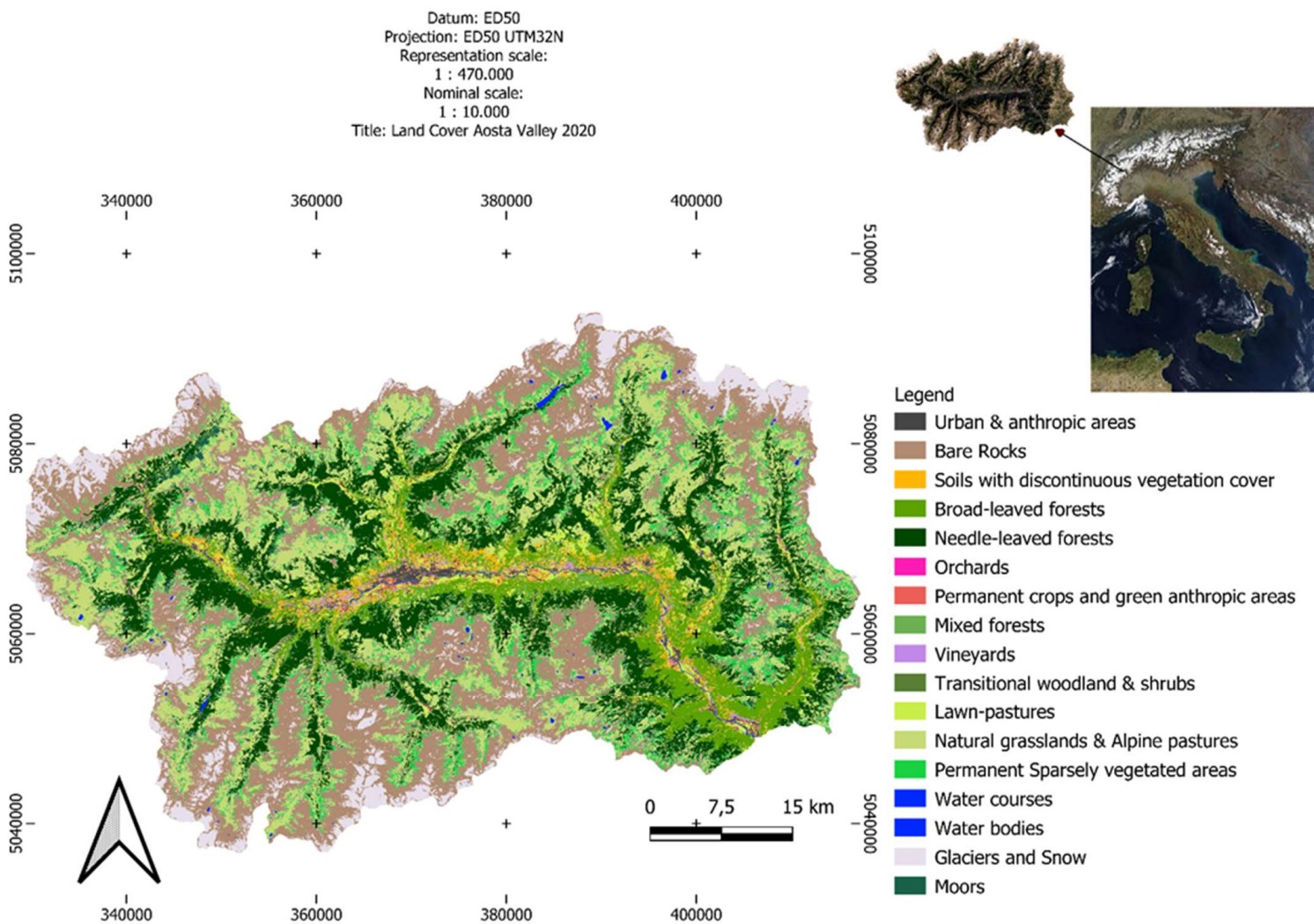

**Figure 8.** Aosta Valley EAGLE Land Cover according to the methodology proposed.

## 5. Discussions

The new EAGLE classes were defined by the European Union (EU) and supported by the European Environmental Agency (EEA) and all the national environmental agencies (in Italy represented by the Istituto Superiore per la Protezione e Ricerca Ambientale—ISPRA). The new classes adopted are more linked to land cover than those presented in Corine Land Cover (CLC) and will be mandatory for regional and national to create LC higher resolution maps starting from the end of 2022 (https://land.copernicus.eu/global/products/lc last accessed on 6 November 2022). In fact, it is worth noting that the previous CLC products had a strong overlap between land cover and land use, generating confusion both in technics and users [16,106]. It is well known that remote sensing permits to define only land cover and sometimes a few numbers of use such as mowing activities [19,107,108]. To map the land use is necessary to ground truth data collected with proximal sensing [108,109]. Following these premises, the new land cover is more linked to the new criteria that consider only biophysical surface components, preserving different levels of detail as the previous CLC products. In https://geoportale.regione.vda.it/download/carta-copertura-suolo/ accessed on 1 September 2022, are reported respectively the legend adopted by ISPRA according to EEA and the final of the present work with a major degree of detail. It is also linked to a detailed explanation of each class. The photo-interpretation correction phase has permitted to increase in the overall accuracy, as well as the K-coefficient by 0.2. Nevertheless, the procedure proposed seems to be sufficiently robust not to require a final phase of refinement and control through photo-interpretation. The combined application of S1 (only for a couple of classes) and S2 EO data seem to boost the classification of land cover components also in mountain areas. Anyway, as suggested in this work, S1 data have to be adopted only to map in addition to optic data urban and water areas because differently a misleading classification may occur due to the physical limits of SAR in mountain areas.

The methodology proposed is scalable to other morphological complex realities, in particular mountainous areas. The tentative approach adopted with free EO data and open-source tools can represent a possible ordinary workflow that can be pursued by territorial planning and management enterprises and agencies to monitor land cover changes through the years and perform new maps. We believe that this approach may have an important technology transfer in mountain territories. This would also help the implementation of local policies concerning new forms of redistribution of contributions with a view to sustainable development. In this regard, each year, Aosta Valley, similar to many Italian regions, has to assign development funds to each municipality, which are largely based on the distribution and extension of the land cover components within its borders. The current method adopted without Remote Sensing (RS), which is extremely dated, provides for sample inspections and GIS analysis using buffer areas that are not representative of reality and which are overlooked by elements deemed of particular interest at a European level. Therefore, this RS methodology may represent a useful tool to reach two goals in one. Firstly, regulate a procedure at a local level that provides for the use of the land cover and an annual update for the purpose of distributing local funds. Secondly, respond to the needs of the European Union by making use of its most recent land cover classification system while promoting technology transfer and the culture of environmental monitoring and sustainability in the public administration.

Cloud computing and archived EO data in the GEE have many advantages for large-scale and composite creation or long timeseries mapping, such as monitoring land cover changes. Moreover, it is easy to integrate EO data coming from different missions and textural features to improve classification accuracy. Nevertheless, the approach suggested goes beyond the simple use of GEE. In fact, GEE has many limitations in terms of the capacity to map mountain areas only in this environment. As shown in this work, SAR data can be adopted only by considering and classifying those classes less affected by SAR distortions. Despite of SAR may map in each weather condition, its application in land cover mapping due to distortions makes its single application weak and feasible. Therefore, optical remote sensing is preferable to SAR for mapping mountain areas. However, their

rational and combined use, as demonstrated by the classifications carried out, allows accurate classification of land cover components that are often difficult to map in mountain areas. A hierarchical approach based on optical and SAR data with different classifications of the results obtained seems to be preferable to classifications based on optical or SAR data only or combined but with a single classification. One of the possible reasons is dictated by the fact that the developed method emphasizes the potential offered by each sensor based on the type of coverage to be mapped or by refining it in the case of combined use. In fact, there are components that respond better to classifications based on timeseries stacks rather than single images or that are based on composite images. Naturally, the consistency of the method proposed here is linked to the time span to be mapped. Therefore, a weekly-based approach would be weaker than a monthly- or yearly-based approach because for mapping the long-term coverage of a component, it is better with more observations and, therefore, more data are available.

Concerning the final matrix obtained, it is interesting to note that a hierarchical approach permits maximizing in complex geomorphological areas both the user and producer accuracies and, at the same time, minimizing the omission and commission errors. Moreover, a wise adoption of Sentinel-1 data (considering only those components that are the least affected by geometrical distortions such as water and urban areas), Sentinel-2 composite data per all classes considering median spectral signature, some indexes, and their standard deviation coupled with multitemporal NDVI and NDRE stack for agronomic classes seem to be the best procedure to map EAGLE Land Cover in alpine areas. This will permit the development of interesting Earth Observation services considering the high spatio-temporal resolution of these missions. In fact, these missions may help planners to develop rural mountain areas tools for distributing funds based on variations in coverage or even planning targeted interventions at a local level in marginal contexts or those that are difficult to access on land. It is interesting to note that simpler machine learning algorithms such as MD and KNN permit reaching the highest accuracies in complex areas with respect to others. According to our knowledge, this is due to the fact the kernel window of these algorithms operates in the near space avoiding abrupt changes that normally occur in complex geomorphological areas and that condition the other ones. Nevertheless, more studies have to be performed in order to overcome remote sensing limitations in mountain areas where it is possible so to permit a real and strong technology transfer worldwide.

## 6. Conclusions

Land cover maps are crucial to monitoring and assessing land cover changes and proposing useful, sustainable management and planning policies. It is worth noting that regions and EU countries, starting in 2022, will have to produce and update land cover products according to the new EAGLE guidelines. Free Copernicus data, offered by S1 and S2 missions, may play a great role in land cover mapping. Nevertheless, the exploitation of these kinds of EO data is well known in the literature, but there is still a lack in the development of a robust methodology to map mountain areas (such as the Alps) with a high level of accuracy, according to EAGLE guidelines. Even if ISPRA proposes a combined use of the radar also in the mountain area, the optical data alone is preferable in these contexts, avoiding problems related to the distortions to which the radar is subjected. At the same time, however, using only optical data, if not processed according to temporal criteria to create composites, prevents the entire territory from being mapped due to shadows or clouds. In this regard, this work explored a possible scalable and repeatable methodology for mountain areas that makes predominant use of optical data but also uses radar data for some components, aiming to compensate for native SAR acquisition mode distortions by adopting a mixed hierarchical approach. An evaluation of different algorithms was conducted, and the most performing for geomorphologically complex territories were the Minimum Distance and the K-Nearest Neighbors. It is worth noting that, according to our knowledge, there is a lack of studies concerning a definition of an approach to overcome remote sensing limitations in mountain areas; therefore, this method could help scientists

and experts deal with remote sensing limits and potentiality in alpine areas having a guideline to map land cover in a complex geomorphological area. The main limitations of this research are represented by: radar distortions that can be compensated but never deleted in alpine areas, ground data continuous updating, and sufficiently numerous to train the model. Finally, the biggest issue that still remains open is the discrimination of rocks and built-up areas due to the fact they both have a similar or equal spectral signature. Therefore, to correctly detect is still necessary a DTM in mountain areas and some manual refining to obtain high accuracies. Nevertheless, the approach proposed may help planners detect land cover changes over time on all components, allowing the regional level to address certain management policies and rational use of available funds.

This suggested methodology may help the implementation of European, as well as global and local policies concerning land cover mapping both at a high spatial and temporal resolution to assess land cover changes due to anthropic pressure and climate change and pursuing a sustainable development perspective empowering the technological transfer in mountainous realities trying to overcome remote sensing limitations that normally is present in alpine areas.

**Author Contributions:** Conceptualization, T.O.; methodology, T.O.; software: T.O. and D.C.; validation, T.O. and D.C.; formal analysis, T.O.; investigation, T.O.; resources, T.O.; data curation, T.O.; writing—original draft preparation, T.O.; writing—review and editing, T.O. and E.B.M.; visualization, T.O. and D.C.; supervision, E.B.M.; project administration, T.O.; funding acquisition, E.B.M. All authors have read and agreed to the published version of the manuscript.

**Funding:** This research received no external funding.

**Institutional Review Board Statement:** Not applicable.

**Informed Consent Statement:** Not applicable.

**Data Availability Statement:** The findings may be reachable at https://geoportale.regione.vda.it/download/carta-copertura-suolo/ (last accessed on 6 November 1993).

**Acknowledgments:** Thanks to the colleagues of INVA spa and GEO4Agri DISAFA Laboratory as well as Annalisa Viani for the support in performing the following work. A remarkable thanks to Edoardo Cremonese for the great feedback regarding this product as well Fabrizia Joly, both of the Environmental Protection Agency of Aosta Valley and Luca Congedo, Michele Munafò and Ines Marinosci of ISPRA Land Unit. A huge thanks to the Regione Autonoma Valle d'Aosta and to the head of GIS area Davide Freppaz and to the head of Cartographic Office Chantal Tréves of the Aosta Valley Region to have permitted to realize this work.

**Conflicts of Interest:** The authors declare no conflict of interest.

## Appendix A

Here, the legend description is reported:

**Table A1.** Land cover description.

| Class Name | EAGLE-ISPRA Presence | Class Number | Description | Class Name | EAGLE-ISPRA Presence | Class Number | Description |
|---|---|---|---|---|---|---|---|
| Permanent crops and green anthropic areas | YES | 21133 | Surfaces affected by anthropogenic activity. Areas affected by agronomic practices in the sensu lato (from simple mowing to irrigation up to plowing/burglary or other soil conditioning practices in most of the time) and the presence of various crops or ornamental green areas conditioned by anthropogenic activities (such as parks, flower beds, sports areas such as turf of soccer fields or arenas). These areas are of a permanent nature without undergoing changes in the type of coverage that characterizes them in the period of time considered. | Bare Rocks | YES | 121 | Natural surfaces. Areas characterized by the presence of rocks, landslides or poorly powerful but consolidated soils in the process of formation. |
| Urban and anthropic areas | YES | 11 | Surfaces strongly influenced by anthropic activity and characterized by human settlements. These are areas in which there are built structures without distinction on the intended use or under construction, as well as roads, airports, railways, parking lots and any artifact capable of determining a permanent or semi-permanent loss of the soil resource. | Soils with discontinuous vegetation cover | YES | 1221 | Natural or natural-shaped surfaces. Areas characterized by unconsolidated soils with continuous coverage over time as they have reduced annual vegetation or xeric sparse vegetation or poorly managed grassing and with little or no agronomic conditioning practices. This coverage also includes jumps in rock or rubble as long as there are spots of vegetation with the presence of spots of little powerful soils and extremely reduced or absent vegetation. |

**Table A1.** *Cont.*

| Class Name | EAGLE-ISPRA Presence | Class Number | Description | Class Name | EAGLE-ISPRA Presence | Class Number | Description |
|---|---|---|---|---|---|---|---|
| Moors | YES | 41 | Natural surfaces. These are areas characterized by an herbaceous-shrubby vegetable association that characterizes slopes and wetlands with usually acid soils, generally cold and humid but well drained and usually poor in humus. The vegetation is mainly made up of Ericaceae (in particular *Calluna vulgaris* L., known as heather from which the term moor or moor derives), *Fabaceae* (such as *Cytisus scoparius* L. in sunnier areas) and junipers (*Juniperus* spp.) | Permanent sparsely vegetated areas | YES | 222 | Natural surfaces. Areas characterized by the presence of areas with scarce but permanent vegetation that is difficult to graze given both the characteristics of the vegetation and in some cases the slope. These are high-altitude surfaces near rocks or natural grasslands and woods. |
| Transitional woodland and shrubs | YES | 21221 | Natural or natural-shaped surfaces. Areas characterized by arboreal species and generally sparse woods near grazing areas or areas with reduced herbaceous vegetation and rocks (such as rubble). These areas indicate dynamics of ecological forest succession following the abandonment of grazing areas and consequent expansion of forest areas or following disturbances to natural or anthropogenic disturbances to the forest. | Vineyards | YES | 21211 | Surfaces influenced by human activity and agronomic practices. Areas characterized by the presence of various cultivation systems of the vineyard. |
| Water bodies | YES | 312 | Natural or natural-shaped surfaces. Areas characterized by the presence of bodies of water such as natural lakes of fluvial and/or glacial origin, artificial reservoirs for the collection and interception of water in correspondence with dams, fishing basins or any other surface of water for recreational or anthropic use. | Broad-leaved forests | YES | 2111 | Natural or natural-shaped surfaces. Wooded areas characterized by a prevalent and widespread presence of broad-leaved trees or broad-leaved species on a given surface (oak, chestnut, ash, maple, lime, alder, birch, poplars, etc.) |

**Table A1.** *Cont.*

| Class Name | EAGLE-ISPRA Presence | Class Number | Description | Class Name | EAGLE-ISPRA Presence | Class Number | Description |
|---|---|---|---|---|---|---|---|
| Water courses | NO | 311 | Natural or natural-shaped surfaces. Areas characterized by the presence of waterways such as rivers, streams, ru and works of hydraulic derivation along runoff lines and slope impluviums. | Needle-leaved forests | YES | 2112 | Natural or natural-shaped surfaces. Wooded areas characterized by a prevalent and widespread presence of conifers or needle-like species on a given surface (larch, spruce, fir, pine, Douglas fir...) |
| Glaciers and snow | YES | 32 | Natural surfaces. Areas characterized by the presence of glaciers, seracs, icefalls and frozen or snow-covered surfaces such as snowfields in the observation period considered. It should be noted how the measurements carried out fall within the full ablation season and can therefore constitute a useful data on the perimeter in this sense. The rock glaciers being totally covered by debris and rocks are not included in this class since they follow a criterion of spectral uniformity of both optical and SAR remote sensing data and therefore refer to the rock class. | Mixed forests | NO | 2114 | Natural or natural-shaped surfaces. Wooded areas characterized by a concomitant presence of broad-leaved trees and conifers. |
| Natural grasslands and alpine pastures | NO | 22112 | Natural or natural-shaped surfaces. Areas characterized by a natural evolution or at most by management conditioning practices at a pastoral level. These areas are characterized by the presence of herbaceous species of medium-high altitude sometimes in correspondence with SPAs, SIC or SACs and of particular naturalistic interest as for some forest areas. In the presence of these surfaces, it is possible to witness grazing activities and the presence of mayen (mountain pastures) with a high historical-cultural and landscape value. | Lawn-pastures Orchards | YES YES | 22111 21131 | Natural-shaped surfaces. Areas characterized by herbaceous cover conditioned by pastoral and agronomic practices in this case mowing, haymaking, and possible irrigation for most of the time. The areas can be characterized by both grazing and mowing. Surfaces affected by human activity and agronomic practices. Areas affected by the presence of orchards or fruit plants for both productive and ornamental purposes. |

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
