# Peer review of "A Possible Land Cover EAGLE Approach to Overcome Remote Sensing Limitations in the Alps Based on Sentinel-1 and Sentinel-2: The Case of Aosta Valley (NW Italy)"

_remotesensing, doi:10.3390/rs15010178_

Round 1

Reviewer 1 Report

I found the manuscript to be interesting and good an example of using land cover EAGLE approach to overcome remote sensing limitations in the Alps based on Sentinel-1 & Sentinel-2 data. The EAGLE is a framework for the integration of land cover and land use (LC/LU) information from various data sets in one single data model. In this study a tentative approach to map land cover overcoming remote sensing limitations in mountains according to the newest EAGLE guidelines was proposed. Results showed that K-Nearest-Neighbor and Minimum Distance classification permit to maximize the accuracy and reduce the errors. a mixed hierarchical approach seems to be the best solution to create LC in mountain areas and strengthen local environmental modelling concerning land cover mapping. I find that the introduction provides decent background, the research design is appropriate, the methods adequately described, and results are clearly presented, so I do not have many major comments. I believe a minor level of revisions should be made to the paper before it is ready to be considered for publication with Remote Sensing.

Specific comment:

One point that could perhaps be strengthened is more of an indication to the readers of the level of uniqueness of the study in introduction section or conclusion: what is main improvement of this study if there is any previous studies in the this topic? 

Author Response

Response to Reviewer 1 Comments

We would like to thank reviewers for their appropriate comments and helpful suggestions that have been carefully considered. Majority of provided suggestions highlighted gaps in the text and were really useful to improve, we hope, paper quality. In blue referees can find their comments, in red authors’ actions to reply/satisfy requests.

In particular, the synthesis of reviewers’ comments suggested a deep revision in paper organization and harmonization. Consequently, you will find some structural changes aimed at simplifying paper reading and make content more effective.

All comments were carefully evaluated and for the most of them corrections and integrations have been provided. Thank you so much for your work!

Point 1 : I found the manuscript to be interesting and good an example of using land cover EAGLE approach to overcome remote sensing limitations in the Alps based on Sentinel-1 & Sentinel-2 data. The EAGLE is a framework for the integration of land cover and land use (LC/LU) information from various data sets in one single data model. In this study a tentative approach to map land cover overcoming remote sensing limitations in mountains according to the newest EAGLE guidelines was proposed. Results showed that K-Nearest-Neighbor and Minimum Distance classification permit to maximize the accuracy and reduce the errors. a mixed hierarchical approach seems to be the best solution to create LC in mountain areas and strengthen local environmental modelling concerning land cover mapping. I find that the introduction provides decent background, the research design is appropriate, the methods adequately described, and results are clearly presented, so I do not have many major comments. I believe a minor level of revisions should be made to the paper before it is ready to be considered for publication with Remote Sensing.

Specific comment:

One point that could perhaps be strengthened is more of an indication to the readers of the level of uniqueness of the study in introduction section or conclusion: what is main improvement of this study if there is any previous studies in the this topic? 

Response 1: Firstly, we would like to thanks the reviewer for his/her kind suggestion and comments. A general comprehensive editing has been done taking into account also the suggestion proposed by the other reviewers.  Concerning the specific comment the referee is right the study proposed according to our knowledge is the first one and unique and we have remaked it as follow: ““It is worth noting that, according to our knowledge there is a lack of studies concerning a definition of an approach to overcome Remote Sensing limitations in the mountain areas, therefore this method could help scientists and experts to deal with remote sensing limits and potentiality in alpine areas having a guideline to map land cover in geomorphological complex area. The main limitations of this research are represented by: radar distortions that can be compensated but never deleted in alpine areas, ground data continuous updating and sufficiently numerous to train the model. Finally, the biggest issue, that still remain open is the discrimination of rocks and built-up areas due to the fact they both have a similar or equal spectral signature. Therefore, to correctly detect is still necessary a DTM in mountain areas and some manual refining to get high accuracies.”

Reviewer 2 Report

The paper deals with an interesting theme highly topical in the context of the current World research trends. Land cover maps are crucial to environmental modelling and define sustainable management and planning policies. The development of a land cover mapping continuous service according to the new EAGLE legend criteria has become of great interest to the public sector. In this work a tentative approach to map land cover overcoming remote sensing limitations in mountains according to the newest EAGLE guidelines was proposed. To reach this goal, the methodology has been developed in Aosta Valley, NW of Italy due to its higher degree of geomorphological complexity. The main aim of this study work has been to develop a strong approach to map Land Cover in geomorphological complex areas scalable to all mountain realities going beyond EO Data limits and produce land cover map with the highest accuracy adopting the most suitable algorithms.

I appreciate especially the detailed map outputs, as well as the interpretation of the obtained results. In order to meet the objective of the paper, the author chose an adequate methodical apparatus based on the use of relevant data and modern geoinformatics equipment.

The title of the paper is acceptable and adequate and no changes are necessary. I find the abstract acceptable and well structured. The manuscript has a sufficient scientific value and the information provided represents widening of knowledge. The conclusions are based entirely on the results and the methods used are adequate. The relation between the scientific value and the extent is acceptable. The language and style of the text are at an acceptable level. The tables and illustrations used in the paper are adequate; however I consider the number of references incomplete. The topic dealt with in the paper is also covered by other authors in papers.

I recommend language correction of the text by a native speaker, if possible. I have no other remarks of a rather significant nature concerning the paper. The results are valuable and the scientific paper brings new original data. The manuscript is acceptable after minor revision with minor amendments required; no re-review is necessary. I recommend the paper for the print.

So no elements that should be corrected:

Conclusion: any limitation of your research? So please add it.

I recommend amending the references. This issue is also covered by the newer papers from other authors. I recommend adding some papers into the references.

Table 4 and 9 – text is very small /little poor quality

As you see, there is not too much to correct according to my opinion.

Good luck in your future scientific work.

Author Response

Response to Reviewer 2 Comments

We would like to thank reviewers for their appropriate comments and helpful suggestions that have been carefully considered. Majority of provided suggestions highlighted gaps in the text and were really useful to improve, we hope, paper quality. In blue referees can find their comments, in red authors’ actions to reply/satisfy requests.

In particular, the synthesis of reviewers’ comments suggested a deep revision in paper organization and harmonization. Consequently, you will find some structural changes aimed at simplifying paper reading and make content more effective.

All comments were carefully evaluated and for the most of them corrections and integrations have been provided. Thank you so much for your work!

Point 1 : The paper deals with an interesting theme highly topical in the context of the current World research trends. Land cover maps are crucial to environmental modelling and define sustainable management and planning policies. The development of a land cover mapping continuous service according to the new EAGLE legend criteria has become of great interest to the public sector. In this work a tentative approach to map land cover overcoming remote sensing limitations in mountains according to the newest EAGLE guidelines was proposed. To reach this goal, the methodology has been developed in Aosta Valley, NW of Italy due to its higher degree of geomorphological complexity. The main aim of this study work has been to develop a strong approach to map Land Cover in geomorphological complex areas scalable to all mountain realities going beyond EO Data limits and produce land cover map with the highest accuracy adopting the most suitable algorithms.

I appreciate especially the detailed map outputs, as well as the interpretation of the obtained results. In order to meet the objective of the paper, the author chose an adequate methodical apparatus based on the use of relevant data and modern geoinformatics equipment.

The title of the paper is acceptable and adequate and no changes are necessary. I find the abstract acceptable and well structured. The manuscript has a sufficient scientific value and the information provided represents widening of knowledge. The conclusions are based entirely on the results and the methods used are adequate. The relation between the scientific value and the extent is acceptable. The language and style of the text are at an acceptable level. The tables and illustrations used in the paper are adequate; however I consider the number of references incomplete. The topic dealt with in the paper is also covered by other authors in papers.

I recommend language correction of the text by a native speaker, if possible. I have no other remarks of a rather significant nature concerning the paper. The results are valuable and the scientific paper brings new original data. The manuscript is acceptable after minor revision with minor amendments required; no re-review is necessary. I recommend the paper for the print.

So no elements that should be corrected:

Conclusion: any limitation of your research? So please add it.

I recommend amending the references. This issue is also covered by the newer papers from other authors. I recommend adding some papers into the references.

Table 4 and 9 – text is very small /little poor quality

As you see, there is not too much to correct according to my opinion.

Good luck in your future scientific work.

Response 1: Firstly, we would like to thanks the reviewer for his/her kind suggestion and comments. A general comprehensive editing has been done taking into account also the suggestion proposed by the other reviewers.  Concerning the specific comment the referee is right some specific references have been added (please see the revised manuscript). Moreover, language correction have been performed. Then table 4 and 9 have been improved in quality.

Finally, in the conclusion section it has been reported limitation of the reseach as follow. “It is worth noting that, according to our knowledge there is a lack of studies concerning a definition of an approach to overcome Remote Sensing limitations in the mountain areas, therefore this method could help scientists and experts to deal with remote sensing limits and potentiality in alpine areas having a guideline to map land cover in geomorphological complex area. The main limitations of this research are represented by: radar distortions that can be compensated but never deleted in alpine areas, ground data continuous updating and sufficiently numerous to train the model. Finally, the biggest issue, that still remain open is the discrimination of rocks and built-up areas due to the fact they both have a similar or equal spectral signature. Therefore, to correctly detect is still necessary a DTM in mountain areas and some manual refining to get high accuracies.”

Reviewer 3 Report

Introduction is well explained and need a little systematic presentation of the interesting ideas, together with some comments on the published approaches in Alpine and mountain areas, integrating Sentinel 1 and 2 images, for example find a contribution at https://www.tandfonline.com/doi/pdf/10.1080/22797254.2017.1365570

Malinowski, R.; LewiÅ„ski, S.; Rybicki, M.; Gromny, E.; Jenerowicz, M.; KrupiÅ„ski, M.; Nowakowski, A.; Wojtkowski, C.; KrupiÅ„ski, M.; Krätzschmar, E.; Schauer, P. Automated Production of a Land Cover/Use Map of Europe Based on Sentinel-2 Imagery. Remote Sens. 2020, 12, 3523. https://doi.org/10.3390/rs12213523

Clerici et al. (2017) see https://doi.org/10.1080/17445647.2017.1372316

I appreciate you fixed the main issues of this topic in Alpine context, the readers and Copernicus data users need to know in developing their reseach and technological remote sensing based developments.

Paper aim/scope and objectives is easy to be understand but it can be presented a little bit more rigurous in order to extract the main ideas you followed.

Materials and methods 

Figure 1 - satellite map must be more bright (use a better contrast setting) and must provide basic geographical names like main rivers, peaks and few town names in order to explain its geographical features. Scale bar units need to be only few with divisions of 10 km. It is not necessary to add a grid. Map of Italy is difficult to be red and does not contain the limits of the regions to use it in explaining the geographical setting. Update the display resolution of the map please.

When explaining the difficulties of the regional context in LCLU mapping it can be easier to provide a detail or more to show the specific issues like for example samples with classes with similar signatures, before and after corrections etc. This can be shown for radar and multispectral imagery as well (ex. a glacial melting area and the transition to pastures, timberlines and forest zones, urban area transition to floodplain etc.). This can be done by providing little complementary samples from imagery.

When explaining satellite image data employed it can be better to avoid the general issues, but to extract some specific features for current imagery as you done already. A synthetic table can be used for image data comparison.

If some terrain data acquition is done (ex. GNSS survey of training areas) you can explain a little more and provide an image (an idea).

Methods - in order to obtain a synthetic view over the methodology it is much more easier to provide a general scheme of the analysis, although it can be a complex one. The paper can have a didactic relevance and this workflow chart could be helpful.

Figure 2 need a display to a higher resolution.

Usually, the followed processing steps need to be explained in terms of relevance for the newly resampled data grids, while technical details like software intruments/menus/commands or parameters need to be explained only if it is necessary in particular image transformations.

Table 3 - please mention the names of columns in table (corect is temporal distances)...

Table 4 is in fact a figure to be integrated with the entire processing scheme and later detailed on a topic. This is in fact a part of a radar data processing tutorial (general aspects, adapted). It is important to focus on your very contribution (partly done).

Binary mapping of urban and water classes can be better explained, includin some samplings of imagery and results.

Sentinel 2 MSI data processing (section 3.4) is well explained and some sample can be helpful. Table 5 could contain a short explanation about the useful data to be extracted in classification (description adapted to the regional context - an idea).

Indices can be presented and explained in a table, providing a column with a short description of their significance in the context of the approach.

OBIA approach for ROI selection together with supervised classification are explained but more technical. It is necessary to extract the specific features and argue more clearly the selection of algorithms and parameters.

Figure 3 is general, it has no explanation/key and can be replaced with a detailed map where you can show on a sample the segmentation result, and the selection of training polygons, and, if possible some points collected from terrains surveying. In the background a sample of image can be useful.

Results - the difference between your product and the CLC data can be explained in discussion section when calibrating/comparing the data output and their significance (map samples can be helpful). It is an essential part of this type of approach.

Again, I think a general scheme of the approach can be helpful. Testing different classification algorithms and their statistical evaluation can be done after the presentation (first description, second interpretation) of the product you obtained after such a complex analythical workflow.

Results need to be listed in a more rigurous order - classification results, classification maps, land cover datasets, statistics, accuracy parameters etc.

The discrimination between classes is a real challenge and need to be explained in more detail by providing some samples where the problem is solved. This is a part of your contribution...

Figure 4 need a better place and some detailed samples with data and map. 

Table 9 is in fact a part of validation which is a topic for the discussion section (can be created separately from results). 

The interpretation of results is not enough. You need to provide it in a separate section of discussions. 

I suggest to open much more the approach to the field situation and to the practical integration of the product in regional resource management and local development. There are some interesting examples but a little more can be explained. For example, you could provide some field photos in continuity and illustrate some practical issues from the region the ESA Copernicus EO product can help in finding practical solutions).

Classes from figure 4 are essential. They were described in appendix but a synthetic table can be helpful in the discussion section as a part of the interpretation of results. Samples of map can help the explanation/description.

The relationship between image features like resolution (spatial, spectral, temporal, even radiometric) and scale of the final products can be explain in this section.

A comparison with similar results from literature can be more visible.

Conclusion - general statements can be integrated with the degree the objectives were fulfilled and with the potential issues in practical interdisciplinary approach in regional context.

References - the list can be updated with the Sentinel 1 and 2 LCLU mapping published paper after 2015 (partly done).

Appendix - is so big section, you can select the most relevant tables and statistics you consider essential for explanation. Class description can be accompanied with some photos/orthophotos samples if you consider is helpful.

Author Response

Response to Reviewer 3 Comments

We would like to thank reviewers for their appropriate comments and helpful suggestions that have been carefully considered. Majority of provided suggestions highlighted gaps in the text and were really useful to improve, we hope, paper quality. In blue referees can find their comments, in red authors’ actions to reply/satisfy requests.

In particular, the synthesis of reviewers’ comments suggested a deep revision in paper organization and harmonization. Consequently, you will find some structural changes aimed at simplifying paper reading and make content more effective.

All comments were carefully evaluated and for the most of them corrections and integrations have been provided. Thank you so much for your work!

Point 1 : Introduction is well explained and need a little systematic presentation of the interesting ideas, together with some comments on the published approaches in Alpine and mountain areas, integrating Sentinel 1 and 2 images, for example find a contribution at https://www.tandfonline.com/doi/pdf/10.1080/22797254.2017.1365570

Malinowski, R.; LewiÅ„ski, S.; Rybicki, M.; Gromny, E.; Jenerowicz, M.; KrupiÅ„ski, M.; Nowakowski, A.; Wojtkowski, C.; KrupiÅ„ski, M.; Krätzschmar, E.; Schauer, P. Automated Production of a Land Cover/Use Map of Europe Based on Sentinel-2 Imagery. Remote Sens. 2020, 12, 3523. https://doi.org/10.3390/rs12213523

Clerici et al. (2017) see https://doi.org/10.1080/17445647.2017.1372316

I appreciate you fixed the main issues of this topic in Alpine context, the readers and Copernicus data users need to know in developing their reseach and technological remote sensing based developments.

Paper aim/scope and objectives is easy to be understand but it can be presented a little bit more rigurous in order to extract the main ideas you followed.

Materials and methods

Figure 1 - satellite map must be more bright (use a better contrast setting) and must provide basic geographical names like main rivers, peaks and few town names in order to explain its geographical features. Scale bar units need to be only few with divisions of 10 km. It is not necessary to add a grid. Map of Italy is difficult to be red and does not contain the limits of the regions to use it in explaining the geographical setting. Update the display resolution of the map please.

When explaining the difficulties of the regional context in LCLU mapping it can be easier to provide a detail or more to show the specific issues like for example samples with classes with similar signatures, before and after corrections etc. This can be shown for radar and multispectral imagery as well (ex. a glacial melting area and the transition to pastures, timberlines and forest zones, urban area transition to floodplain etc.). This can be done by providing little complementary samples from imagery.

When explaining satellite image data employed it can be better to avoid the general issues, but to extract some specific features for current imagery as you done already. A synthetic table can be used for image data comparison.

If some terrain data acquition is done (ex. GNSS survey of training areas) you can explain a little more and provide an image (an idea).

Methods - in order to obtain a synthetic view over the methodology it is much more easier to provide a general scheme of the analysis, although it can be a complex one. The paper can have a didactic relevance and this workflow chart could be helpful.

Figure 2 need a display to a higher resolution.

Usually, the followed processing steps need to be explained in terms of relevance for the newly resampled data grids, while technical details like software intruments/menus/commands or parameters need to be explained only if it is necessary in particular image transformations.

Table 3 - please mention the names of columns in table (corect is temporal distances)...

Table 4 is in fact a figure to be integrated with the entire processing scheme and later detailed on a topic. This is in fact a part of a radar data processing tutorial (general aspects, adapted). It is important to focus on your very contribution (partly done).

Binary mapping of urban and water classes can be better explained, includin some samplings of imagery and results.

Sentinel 2 MSI data processing (section 3.4) is well explained and some sample can be helpful. Table 5 could contain a short explanation about the useful data to be extracted in classification (description adapted to the regional context - an idea).

Indices can be presented and explained in a table, providing a column with a short description of their significance in the context of the approach.

OBIA approach for ROI selection together with supervised classification are explained but more technical. It is necessary to extract the specific features and argue more clearly the selection of algorithms and parameters.

Figure 3 is general, it has no explanation/key and can be replaced with a detailed map where you can show on a sample the segmentation result, and the selection of training polygons, and, if possible some points collected from terrains surveying. In the background a sample of image can be useful.

Results - the difference between your product and the CLC data can be explained in discussion section when calibrating/comparing the data output and their significance (map samples can be helpful). It is an essential part of this type of approach.

Again, I think a general scheme of the approach can be helpful. Testing different classification algorithms and their statistical evaluation can be done after the presentation (first description, second interpretation) of the product you obtained after such a complex analythical workflow.

Results need to be listed in a more rigurous order - classification results, classification maps, land cover datasets, statistics, accuracy parameters etc.

The discrimination between classes is a real challenge and need to be explained in more detail by providing some samples where the problem is solved. This is a part of your contribution...

Figure 4 need a better place and some detailed samples with data and map.

Table 9 is in fact a part of validation which is a topic for the discussion section (can be created separately from results).

The interpretation of results is not enough. You need to provide it in a separate section of discussions.

I suggest to open much more the approach to the field situation and to the practical integration of the product in regional resource management and local development. There are some interesting examples but a little more can be explained. For example, you could provide some field photos in continuity and illustrate some practical issues from the region the ESA Copernicus EO product can help in finding practical solutions).

Classes from figure 4 are essential. They were described in appendix but a synthetic table can be helpful in the discussion section as a part of the interpretation of results. Samples of map can help the explanation/description.

The relationship between image features like resolution (spatial, spectral, temporal, even radiometric) and scale of the final products can be explain in this section.

A comparison with similar results from literature can be more visible.

Conclusion - general statements can be integrated with the degree the objectives were fulfilled and with the potential issues in practical interdisciplinary approach in regional context.

References - the list can be updated with the Sentinel 1 and 2 LCLU mapping published paper after 2015 (partly done).

 Appendix - is so big section, you can select the most relevant tables and statistics you consider essential for explanation. Class description can be accompanied with some photos/orthophotos samples if you consider is helpful.

Response 1: Firstly, we would like to thanks the reviewer for his/her suggestion and comments. A general comprehensive editing has been done taking into account also the suggestion proposed by the other reviewers therefore not all the several point proposed has been changed.  Concerning the specific comments:

In the introduction

In order to enrich and give a systematic overview of the analyzed topic some papers have been included into the manuscript also some of those adviced. Please see into the introduction section. In particular it was added “In particular, many studies have adopted Sentinel-1 or Sentinel-2 to map land cover and other land cover components in some alpine areas but nobody on the limits and potential-ities offered by a coupled adoption of SAR and multispectral data in mountain areas [53–56].”

Concerning the commenti “I appreciate you fixed the main issues of this topic in Alpine context, the readers and Copernicus data users need to know in developing their reseach and technological remote sensing based developments.” We thank the reviewer and following the advice on paper aim/scope and objectives it has reformulated has follow: “Finally, in order to achive the main goal of this work, it has been to developdeveloped a scalable worldwide approach capable of mapping EAGLE Land Cover in mountain areas with higher accuracies overcoming Remote Sensing limitations and trying to exploit only the potentialities offered by radar and optical in geomorphological complex areas. In particular, allowing it has been created the creation of a localan alpine alpine suitable op-erative procedure to map and products with high spatial and temporal resolution and update frequency Land Cover for environmental planning and management for opera-tional purposes following the European guidelines of EAGLE. The product realized start-ing from the approach developed is compatible with the old Corine Land Cover and new rules in terms of kind of classes that have to be adopted as well as, creating a continuous service to help alpine region to monitor a huge amount of component of the territory at a higher spatial resolution to help local, national, European and international planners [39,41] for the production of pixel-based land cover classification products [57].”

In order to be more rigurous and extract the main ideas as you asked.

Material and methods

Concerning Fig.1 brightness has been risen up. We have not added city names, rivers etc because of we just want to present the AOI and not mislead the reader we are performing a Remote Sensing analysis not a pure GIS, therefore we have preferred to mantain the one realized. It is worth noting that the bounadary of the region is represented by the zoom image of the Aosta Valley Region.

Concerning the difficulties of the regional context in LCLU mapping we agree with the referee we have better described into the text and not added spectral signature as you kindly suggested because it was not a problem of signature but of algortithm and workflow follow to obtain the higher accuracies. To answer this point we have added the following information. “It is worth noting that, remote sensing limitation in this area, as well, as in mountain areas worldwide can be overcome reasonably adopting a step-by-step hierarchical classi-fication approach (as presented in this work) by detecting which algorithm seems to best map land cover components and exploit the best workflow per each class so as to opti-mize accuracy and reduce error. The question is not so much linked to forms of manual compensation of spectral or geometric signatures but to a careful use of satellite sensors on the basis of the little scientific literature present on this topic in the mountain area and ac-cording to the type of coverage and algorithm tests of machine learning as conducted in this study in order to define a scalable approach that allows a technology transfer.”

Concerning satellite explanation tables of bands adopted per each satellite have ben reported. Please see into the manuscript.

Concerning the GNSS points as suggested by the reviewer an image was added and a short detail describtion as it follows: The GNSS data were acquired into the Aosta Valley region in well-known classes. The boundaries were defined through perimeter detection or a-posteriori through photo-interpretation of Planet images. Here below, we report an overview.”

In methods

As suggested by the reviewer a synthetic view over the methodology was realized in order to help others researchers and improve the didactic relevance of the manuscript. Plese see into the manuscript.

The referee is right the image has been changed. Please see into the manuscript.

Concerning the software intruments/menus/commands or parameters explained this is done in order to permit research scalability which is one of the most important topic in open access science. We agree with the reviewer that only cruicial part have to be written anyway as he/she suggests in a previous part the manuscript has to assume also a didactic relevance therfore to be clear in each step is crucial for us.

Concerning table 3 the referee is right the correction has been done please see into the text.

Concerning table 4 we partialy agree with the referee, anyway we leave the radar processing workflow because it is crucial in the suggested methodology to obtain in alpine urban areas mask.

Concerning binary mapping of urban and water classes, as suggested by the reviewer, has better explained as follow: “Urban and water masks were adopted in the following phase using optical data to better refine this classes which are the unique less affected by SAR distortions in mountain areas. The refining between SAR and optical data were performed manually.” We have not included the images because the article can be too long and the mask has been used as explained in a complex workflow.

Concerning Sentinel 2 MSI data processing (section 3.4) we thank the reviewer and we have reformulated thi section with a particular focus on table 5. Please see the caption section into the text.

Concerning the indices we totally agree with the reviewer. Please see the new table into the text.

Concerning the OBIA we agree with the referee therefore this section has been better explained.

Concerning Figure 3 the referee is partialy right. Therefore, following his/her suggestion we have added a figure of points detected on the ground for both the training and validation set and we have added an explanation/key in the caption section of figure 3.

Results

Concerning difference between CLC and the map realized we agree with the referee and we have added a discussion section in order to better organize the paper and taking into account his/her suggestion plese see into the text the discussion section.

 Concerning the results description the referee is right. Firstly, as suggested we have provied a general scheme of the approach then a more rigurous order description was done. Please see into the text.

Concerning the discrimination between classes we agree with the reviewer therefore we have better explained as follow To better discriminate these classes it was adopted the above-mentioned procedure because a simple classification with only median spectral sign and median indexes are not able to carefully detect agronomic cultivation as demonstrated by [10]. NDVI and NDRE temporal stack were considered including textural pattern in order to carefully map these classes considering that they are very complex to detect especially in mountain areas. This procedure has permitted to rise up the user accuracy for both classes of 27% reaching per each class mapped a user accuracy more then 90%.”

Concerning figure 4 we agree with the reviewer we have move it. The map can be donwload at the link provide in supplementary material.

Concerining the confusion matrix it was moved just before the discussion section and in this part discussed as suggested. In fact a discussion section has been created please see into the text.

Concering to enrich the discussion we agree with the reviewer anyway we have not provided more images because the article in the present form is long i hope the reviewer will understand.

Concerning classes from figure 4 we agree with the reviewer and we have added the in the discussion section.

Concerning the relationship between image features like resolution (spatial, spectral, temporal, even radiometric) and scale of the final products has been explained in the discussion section.

Concerning the comparison with similar results from literature and objectives and potential issues in practical interdisciplinary approach in regional context the reviewer is right we have better remarke i tinto the conclusion section as it follow: “It is worth noting that, according to our knowledge there is a lack of studies concerning a definition of an approach to overcome Remote Sensing limitations in the mountain areas, therefore this method could help scientists and experts to deal with remote sensing limits and potentiality in alpine areas having a guideline to map land cover in geomorphological complex area. The main limitations of this research are represented by: radar distortions that can be compensated but never deleted in alpine areas, ground data continuous updating and sufficiently numerous to train the model. Finally, the biggest issue, that still remain open is the discrimination of rocks and built-up areas due to the fact they both have a similar or equal spectral signature. Therefore, to correctly detect is still necessary a DTM in mountain areas and some manual refining to get high accuracies. Nevertheless, the approach proposed may help planners in detect land cover changes over time on all components, allowing regional level to address certain management policies and rational use of available funds.

Concerning references referee is right we have added citations and maintain the older one also.

Concerning Appendix A we consider it all exential therefore we have let it.

Round 2

Reviewer 3 Report

Figure 1 - map based on satellite image can be more brigher and some geographical names like rivers, peaks and towns can help the reader.

Table 4 is in fact a scheme or a figure. Please adapt it. If there are only general aspects regarding interferometric data processing you need to provide an explanation on the specific issues for your study/approach.

I really appreciate the didactic component of your approach. 

Figure 4 can be presented with a legend of the colors you employed. A detail map of ROI pattern can be better used instead of this general map. It can be interesting to show it and explain only the most problematic aspects in samples selection and mapping (ex. an area with a high variety of land cover classes and vegetation zones).

Page 25 - table title EAGLE is correct.

I recommend very much to provide some detailed map sections to a special figure in discussion section. One or two samples from totally different areas can be helpful to identify the agreement between the results and the ground truth features (some field photos could be also useful).

Author Response

Response to Reviewer 3 Comments

We would like to thank reviewers for their appropriate comments and helpful suggestions that have been carefully considered. Majority of provided suggestions highlighted gaps in the text and were really useful to improve, we hope, paper quality. In blue referees can find their comments, in red authors’ actions to reply/satisfy requests.

In particular, the synthesis of reviewers’ comments suggested a deep revision in paper organization and harmonization. Consequently, you will find some structural changes aimed at simplifying paper reading and make content more effective.

All comments were carefully evaluated and for the most of them corrections and integrations have been provided. Thank you so much for your work!

Point 1 : Figure 1 - map based on satellite image can be more brigher and some geographical names like rivers, peaks and towns can help the reader.

Table 4 is in fact a scheme or a figure. Please adapt it. If there are only general aspects regarding interferometric data processing you need to provide an explanation on the specific issues for your study/approach.

I really appreciate the didactic component of your approach.

Figure 4 can be presented with a legend of the colors you employed. A detail map of ROI pattern can be better used instead of this general map. It can be interesting to show it and explain only the most problematic aspects in samples selection and mapping (ex. an area with a high variety of land cover classes and vegetation zones).

Page 25 - table title EAGLE is correct.

I recommend very much to provide some detailed map sections to a special figure in discussion section. One or two samples from totally different areas can be helpful to identify the agreement between the results and the ground truth features (some field photos could be also useful).

Response 1: Firstly, we would like to thanks the reviewer for his/her suggestion and comments.

Concerning Figure 1 has been changed taking into account the suggestion. Please see into the text.

Concerning Table 4 is a preocedure to correctly calibrate the SAR images in SNAP. It is necessary to permit to scale this research in other mountain areas and moreover it has a didactic relevance. Anyway, this has been added into the text to better specify: “The processing steps to correctly calibrate SAR images in order to map and discriminate urban and not urban areas have been reported here below. It is worth noting that these steps have been realized in the ESA SNAP v.8.0.0 toolbox (see more detail in table 4).

Concerning the didactic approach thank you so much.

Concerning Figure 4 the referee is right we have employed a color legend. Concerning the ROI pattern, they are really small in some places and due to the wide spatial extent is difficult to represent in images. Anyway, we have explained into the text that the distribution of ROI has follow precise rules. Please look at: 3.7 section.

Concerning page 25 yes, it is.

Concerning detailed map in the discussion we agree with the reviewer and we have added a new figure. Please see into the text.
